# Self-supervised Adversarial Purification for Graph Neural Networks

**Woohyun Lee** [1]  **Hogun Park** [1]

## Abstract

Defending Graph Neural Networks (GNNs) against adversarial attacks requires balancing accuracy and robustness, a trade-off often mishandled by traditional methods like adversarial training that intertwine these conflicting objectives within a single classifier. To overcome this limitation, we propose a self-supervised adversarial purification framework. We separate robustness from the classifier by introducing a dedicated purifier, which cleanses the input data before classification. In contrast to prior adversarial purification methods, we propose GPR-GAE, a novel graph auto-encoder (GAE), as a specialized purifier trained with a self-supervised strategy, adapting to diverse graph structures in a data-driven manner. Utilizing multiple Generalized PageRank (GPR) filters, GPR-GAE captures diverse structural representations for robust and effective purification. Our multi-step purification process further facilitates GPR-GAE to achieve precise graph recovery and robust defense against structural perturbations. Experiments across diverse datasets and attack scenarios demonstrate the state-of-the-art robustness of GPR-GAE, showcasing it as an independent plug-and-play purifier for GNN classifiers. Our code can be found in https://github.com/woodavid31/GPR-GAE.

## 1. Introduction

Graph Neural Networks (GNNs) have become a powerful tool for learning on graph-structured data, with applications in social networks (Fan et al. 2019), biological networks (Zhang et al. 2021), and recommender systems (Wu et al. 2022). Moreover, they have been expanded to domains such as self-supervised learning (Jung & Park 2025) and explainability (Kang et al. 2024). However, due to their reliance on the underlying graph structure, even small perturbations can drastically degrade their performance, making them highly vulnerable to adversarial attacks (Xu et al. 2019a; Geisler et al. 2021). Defending against such attacks requires balancing accuracy and robustness. Accuracy measures performance on clean data, while robustness reflects resilience to adversarial attacks. Focusing on accuracy can make a model rely on adversarially vulnerable features, while prioritizing robustness may reduce this reliance but harm clean data performance.

Adversarial training (Goodfellow et al. 2015; Madry et al. 2018), one of the earliest defense methods in the image domain, has been adapted into GNNs (Xu et al. 2019a; Feng et al. 2019; Li et al. 2022; Zhang et al. 2022; Gosch et al. 2023) to enhance robustness under adversarial settings in graphs. However, it inherently intertwines the two conflicting objectives in a single learning framework. This limits the achievable robustness, as pursuing higher robustness often comes at the expense of significant accuracy loss. Moreover, adversarial training often falls short in broad applicability, showing efficiency in a narrow set of GNNs.

On the other hand, adversarial purification (Wu et al., 2019; Entezari et al., 2020; Li et al., 2024) provides a viable framework by introducing a separate module—a purifier—that preprocesses the input data to defend against attacks. It focuses on removing adversarial components and purifying the input data, allowing each module to specialize in its own role: the purifier enhances robustness, while the classifier focuses solely on accuracy. This separation provides clear pathways for each module to leverage its strengths independently, avoiding interference between the conflicting objectives of accuracy and robustness. However, existing purification methods do not fully capitalize on this potential, often relying on predefined heuristics without a dedicated training process (Wu et al. 2019; Entezari et al. 2020) or a simple edge detection method (Li et al. 2024). Heuristic-based approaches lack adaptivity and leave defenses highly vulnerable to carefully crafted attacks (Mujkanovic et al. 2022), while classifier-derived edge representations do not capture intricate structural differences between clean and adversarial edges, resulting in suboptimal performance.

Based on these observations, our work makes several key contributions to adversarial purification in GNNs. First, we

[1]Department of Computer Science and Engineering, Sungkyunkwan University, Suwon, South Korea. Correspondence to: Hogun Park <hogunpark@skku.edu>.

*Proceedings of the $42^{nd}$ International Conference on Machine Learning*, Vancouver, Canada. PMLR 267, 2025. Copyright 2025 by the author(s).

analyze and decouple the conflicting objectives of accuracy and robustness in adversarial training into two components: a classifier for accuracy and a purifier trained independently to restore graph structures. Under the adversarial purification framework, we design a self-supervised training strategy, ensuring the purifier's independence from the classification task and mitigating trade-offs. Second, as the purifier, we introduce GPR-GAE, a novel graph auto-encoder architecture that leverages multiple Generalized PageRank (GPR) filters to capture diverse and unique neighborhood representations. This allows effective distinction between clean and adversarial graph regions by modeling complex structural differences, while enabling accurate encoding and reconstruction of a cleaner graph structure. Third, we adopt a multi-step purification process that iteratively refines the graph with a convergent nature, yielding more precise and effective purification. Finally, through extensive experiments across diverse datasets and attack scenarios, we demonstrate that GPR-GAE excels as a specialized purifier, achieving state-of-the-art performance while serving as a plug-and-play defense module that is compatible with various GNNs.

## 1.1. Related Work

**Adversarial Training:** Gosch et al. (2023) emphasizes that flexible GNN architectures (Chien et al. 2021; He et al. 2022) excel in adversarial training settings due to their ability to learn robust propagation pathways by dynamically weighting different powers of the adjacency matrix.

**Robust GNNs:** Some methods forgo adversarial training and focus on designing a robust GNN architecture as a defense. For example, EvenNet (Lei et al. 2022) ignores odd-hop neighbors and uses an even-polynomial graph filter, while SoftMedianGDC (Geisler et al. 2021) uses soft-median aggregation with GDC to mitigate the influence of outliers. However, robust GNNs share the same limitations as adversarial training, where accuracy and robustness goals are tightly coupled within the architectural designs, constrained by inherent trade-offs.

**Adversarial Purification:** Jaccard-GCN (Wu et al. 2019) prunes edges between nodes that have Jaccard similarity below a threshold. SVD-GCN (Entezari et al. 2020) finds that attacks tend to affect high-rank components of the graph and perform a low-rank approximation to purify the graph. GOOD-AT (Li et al. 2024) integrates adversarial training and purification by generating adversarial samples using a PGD attack (Xu et al. 2019a) on a pre-trained GCN (Kipf & Welling 2017) classifier. It encodes edges with features and classifier logit embeddings, training an ensemble of MLP detectors to distinguish between clean (in-distribution) and adversarial (out-of-distribution) edges. During testing, edges flagged as OOD by any detector are pruned.

## 2. Preliminary

### 2.1. Problem Formulation and Notation

We represent a graph $G$ as $G = (\mathcal{V}, \mathcal{E})$, where $\mathcal{V}$ is the set of $N$ nodes and $\mathcal{E}$ is the set of edges, or equivalently as $(\mathbf{A}, \mathbf{X})$, with $\mathbf{A} \in \mathbb{R}^{N \times N}$ as the adjacency matrix and $\mathbf{X} \in \mathbb{R}^{N \times F}$ as the feature matrix, where $F$ denotes the feature dimension. While a normalized adjacency matrix with self-loops, $\tilde{\mathbf{A}}_s = \mathbf{D}_s^{-1/2}(\mathbf{A} + \mathbf{I})\mathbf{D}_s^{-1/2}$ is commonly used, our approach adopts $\tilde{\mathbf{A}}_{ns} = \mathbf{D}^{-1/2}\mathbf{A}\mathbf{D}^{-1/2}$, excluding self-loops. Here, $\mathbf{D}_s$ and $\mathbf{D}$ are the respective degree matrices.

This work addresses adversarial attacks on Graph Neural Networks (GNNs) through structural perturbations, where attackers manipulate edges (insertions or deletions) to degrade performance. We denote the perturbed graph as $G' = (\mathbf{A}', \mathbf{X})$ and seek to recover a structure closer to the clean graph $G = (\mathbf{A}, \mathbf{X})$. Unlike poisoning attacks targeting the training phase, we focus on **evasion attacks**, the same problem setting as adversarial training, which occurs at the test phase. Following Gosch et al. (2023), we employ an **inductive setting**, excluding validation and test nodes during training to prevent information leakage (memorizing the graph for defense) and ensure fair evaluation.

### 2.2. Learning Objective of Adversarial Training

Adversarial training (Xu et al. 2019a; Li et al. 2022; Gosch et al. 2023) enhances the robustness of a GNN classifier $f_\psi$ against evasion attacks by iteratively generating adversarial examples and training the classifier on them. The overall objective can be formulated as:

$$\mathcal{L}(\psi) = \max_{G' \in \mathcal{B}(G, \epsilon)} \mathbb{E}_{v \in \mathcal{V}}[\ell(f_\psi(G', v), y_v)]. \quad (1)$$

Here, $G'$ is an adversarially perturbed graph sampled from $\mathcal{B}(G, \epsilon)$, the set of graphs obtained from $G$ via a bounded number of edge or feature modifications and constrained by the perturbation budget $\epsilon$. Each node $v \in \mathcal{V}$ has a true label $y_v$, and $\ell$ denotes the loss function. Training alternates between generating the worst-case $G'$, which maximizes the loss, and updating classifier parameters $\psi$ to minimize the expected loss over $G'$.

### 2.3. Generalized PageRank Filter

The Generalized PageRank (GPR) filter, introduced in GPRGNN (Chien et al., 2021), provides a flexible mechanism for dynamically adjusting the contribution of each propagation step during message passing. This is achieved through learnable coefficients $\gamma_k$, which weigh the $k$-th power of the normalized adjacency matrix $\tilde{\mathbf{A}}_s^k$. The propagation rule is expressed as $\mathbf{H} = \sum_{k=0}^{K} \gamma_k \left( \tilde{\mathbf{A}}_s^k \mathbf{H}^{(0)} \right)$, where in GPRGNN, $\mathbf{H}^{(0)}$ is the initial feature-driven logit embeddings. By learning the coefficients $\gamma = [\gamma_0, \ldots, \gamma_K]$,

the GPR filter flexibly adapts the contribution of information from different neighborhood ranges, up to $K$-hops. This adaptability allows GPR filters to dynamically capture and aggregate neighborhood information, being suitable for both homophilic and more complex heterophilic graphs.

## 3. Motivations

### 3.1. Self-supervised Adversarial Purification

We discuss the motivations behind our use of a self-supervised adversarial purification framework for defending against adversarial attacks. We first analyze the learning objective of adversarial training and highlight the inherent trade-off between accuracy and robustness. The adversarial training objective can be decomposed as follows:

**Theorem 3.1** (Decomposition of Adversarial Training). *Given disjoint sets of nodes $\mathcal{V}_{unaffected} \subseteq \mathcal{V}$ and $\mathcal{V}_{affected} \subseteq \mathcal{V}$, representing those unaffected and affected by adversarial perturbations in G', respectively, such that $\mathcal{V}_{unaffected} \cup \mathcal{V}_{affected} = \mathcal{V}$, let $\lambda = \frac{|\mathcal{V}_{unaffected}|}{|\mathcal{V}|}$, where $\lambda \in [0, 1]$, denote the proportion of unaffected nodes. The adversarial training objective for a GNN classifier $f_\psi$ can be decomposed as follows:*

$$\mathcal{L}(\psi) = \lambda \cdot \underbrace{\mathbb{E}_{v \in \mathcal{V}_{unaffected}}\big[\ell(f_\psi(G, v), y_v)\big]}_{Accuracy\ Term}$$
$$+ (1 - \lambda) \cdot \max_{G' \in \mathcal{B}(\mathcal{G}, \epsilon)} \underbrace{\mathbb{E}_{v \in \mathcal{V}_{affected}}\big[\ell(f_\psi(G', v), y_v)\big]}_{Robustness\ Term}.$$

*Proof.* Provided in Appendix A.1. □

In particular, a node $v$ is considered affected by an adversarial perturbation if a modification exists within its local propagation boundary. For instance, in a 2-layer GCN (Kipf & Welling, 2017), a node $v$ is affected if there are perturbations within its two-hop neighborhood.

The decomposition separates the adversarial training objective into two distinct terms: accuracy and robustness. Given that the local structure of node $v$ is denoted as $S_v$ and the perturbed local structure as $S_v'$, the accuracy term focuses on learning the mapping from $(v, S_v)$ to the true label $y_v$, while the robustness term focuses on learning the mapping from $(v, S_v')$ to $y_v$. Meanwhile, within the decomposition, the attack budget $\epsilon$ indirectly determines $|\mathcal{V}_{unaffected}|$, controlling the value of $\lambda$ and emphasis on each term.

However, excessive emphasis on the robustness term can introduce perturbations that drastically alter the nodes' semantics, rendering the mapping from $(v, S_v')$ to $y_v$ meaningless for learning useful representations. The coexistence of such potentially detrimental learning within the overall objective can hinder the learning of meaningful representations from the clean graph structure, ultimately constraining

performance on clean data. As a result, most adversarial training methods keep the attack budget during training relatively low to preserve the clean accuracy. Consequently, it inherently limits the level of robustness that can be achieved through training, leaving the classifier vulnerable to stronger attacks that exceed the restricted budget.

The limitations in adversarial training motivates a fundamental redesign: *can we decouple the objectives of accuracy and robustness, assigning them to specialized modules that address each goal more effectively?*

Therefore, we propose a self-supervised adversarial purification framework with clear efforts to fully decouple the accuracy and robustness objectives. The two are explicitly handled and learned in separate modules with distinct specializations. This contrasts with previous adversarial purification approaches, which either replace the purifier with predefined heuristics or make the purifier dependent on the classifier's samples and representations.

The **accuracy objective** is assigned to a standard GNN classifier, $f_\psi$, which focuses solely on learning accurate representations from the clean graph $G$. This is reflected in the following typical supervised learning objective:

$$\mathcal{L}(\psi) = \mathbb{E}_{v \in \mathcal{V}}\big[\ell(f_\psi(G, v), y_v)\big]. \tag{2}$$

The **robustness objective**, in turn, is assigned to a separate purifier model, $f_\theta$, which does not learn to predict node labels. Instead, it focuses on reconstructing the clean graph $G$ from perturbed inputs $G'$, which naturally aligns with the following self-supervised objective:

$$\mathcal{L}(\theta) = \ell(f_\theta(G'), G), \quad G' \sim \mathcal{B}(G, \epsilon). \tag{3}$$

In our self-supervised approach, unlike adversarial training's worst case samples, we randomly choose $G'$ from the sample space $\mathcal{B}$, with a relatively large training budget $\epsilon$. This allows the purifier $f_\theta$ to learn general purification directions across a broader range of perturbations around $G$, enabling more robust and adaptive purifications.

### 3.2. Convergent Multi-Step Purification Framework

Traditional purification methods often use a single-step process, modifying the perturbed graph $G'$ by pruning or retaining edges based on fixed criteria. Although simple, these methods make abrupt changes with rough discretization of the graph, limiting their ability to precisely recover clean graph structures under complex or heavy perturbations.

The proposed **multi-step purification** framework addresses these limitations by iteratively refining the graph through a dynamic, data-driven approach. Instead of abrupt corrections, the purifier model $f_\theta$ progressively adjusts the graph structure using the learned purification directions, adapting

to the graph's current state at each step. This gradual refinement not only enhances recovery accuracy but also enables the method to handle severe perturbations more effectively. Moreover, a convergent nature is crucial in multi-step purification in order to ensure stability and prevent oscillatory or divergent behavior during iterative refinement. The following theoretical analyses formalize the convergent behavior of our self-supervised multi-step purification framework:

**Assumption 3.2.** Under optimal self-supervised training over a sufficiently large and continuous perturbation space $\mathcal{B}(G, \epsilon)$ around the clean graph $G$, the purification model $f_\theta$ behaves as a locally Lipschitz continuous function with Lipschitz constant $L \in [0, 1)$.

The local Lipschitz behavior of $f_\theta$ arises from the self-supervised training objective, which minimizes the reconstruction error $\|f_\theta(G') - G\|$ for perturbed inputs within $\mathcal{B}(G, \epsilon)$ while enforcing $f_\theta(G) = G$ for the clean graph. Under optimal training, the model is encouraged to satisfy $\|f_\theta(G') - f_\theta(G)\| \le p\|G' - G\|$ for some small constant $p \in [0, 1)$, implying that $f_\theta$ approximates a locally Lipschitz continuous mapping with Lipschitz constant $L \in [0, 1)$ around $G$. In the inductive setting, the test graph is semantically similar to $G$, and thus the locally Lipschitz property of $f_\theta$ extends naturally to perturbation neighborhoods around the test graph. Empirical verification of the assumption under various dataset is shown in Section 6.2.

**Theorem 3.3** (Convergence of Multi-Step Purification). *Let $G^{(0)} = G'$ be an adversarially perturbed graph, and let $f_\theta$ be a purification model trained and applied in a self-supervised manner. At each step $t$, define the purification direction as $\Delta G^{(t)} = f_\theta(G^{(t)}) - G^{(t)}$, and update the graph iteratively as $G^{(t+1)} = G^{(t)} + \alpha \cdot \Delta G^{(t)}$, where $\alpha \in (0, 1]$ is the step size. Given Assumption 3.2, the sequence $(G^{(t)})_{t \ge 0}$ converges linearly to a stationary point $G^*$.*

*Proof.* Provided in Appendix A.2. □

The theorem guarantees that, under the local Lipschitz condition induced by the self-supervised training objective, the iterative purification process converges to a stable graph structure regardless of the step size. Importantly, this convergence is governed by the behavior of the purification model in isolation. In contrast, incorporating external signals, such as arbitrary classifier outputs, may interfere with the stability of the refinement process. The proposed decoupling of the purifier from the classifier is therefore essential for ensuring both theoretical convergence and robustness against adversarial attacks.

### 3.3. Designing the Purifier

To effectively differentiate clean from attacked graph regions while adapting to diverse structures, purification meth-

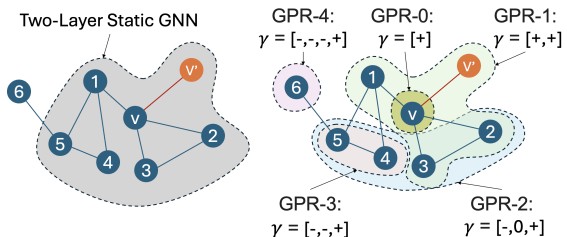

*Figure 1.* **Conceptual Comparison:** Node $v$'s receptive fields (dotted lines) when adversarially linked to node $v'$ (from a different community). *Left:* Two-layer static GNN. *Right:* Five GPR filters, each with unique coefficients $\gamma = [\gamma_0, \ldots, \gamma_K]$ of varying scales. $\gamma_m$ controls neighbors m-hop away from node $v$. In GPR-2, self-suppression ($\gamma_0 < 0$) and two-hop neighborhood emphasis ($\gamma_2 > 0$) form a boundary around nodes $\{2,3,4,5\}$. GPR-3 further suppresses one-hop neighborhood ($\gamma_1 < 0$), excluding nodes 2 and 3, to form a boundary with nodes $\{4,5\}$.

ods must well leverage structural properties such as neighborhood relationships or higher-order dependencies. Graph auto-encoders (GAEs) are naturally suited for this task, as they encode and reconstruct graph structures, modeling both node and edge relationships. However, traditional GAEs face two key challenges in adversarial settings:

**Adversarial Smoothing:** Perturbations in the graph can propagate through the structure, blurring distinctions between clean and attacked regions. Traditional GAEs rely on static GNN layers (Kipf & Welling 2017; Hamilton et al. 2017; Xu et al. 2019b) for neighbor aggregation, which inadvertently smooth local variations. This causes adversarially connected nodes to become similar in representation, reducing the model's ability to differentiate clean and attacked regions. Moreover, standard GAE decoders (Pan et al. 2018; Li et al. 2023) often use proximity-based measures such as inner or element-wise products, further exacerbating this issue by reconstructing spurious connections.

**Difficulties in Higher-Order Representations:** Capturing higher-order structural information in GAEs typically requires stacking multiple GNN layers. However, as the number of layers increases, oversmoothing occurs (Rusch et al. 2023), where node representations across the graph become indistinguishable. This limits traditional GAEs to low-order structural properties, further reducing their efficacy in purifying attacked graphs.

Motivated by these limitations, we propose GPR-GAE, a novel graph auto-encoder architecture that utilizes diverse representations from multiple GPR filters, addressing the challenges of traditional GAEs while enhancing the structural capabilities. The key intuitions are as follows:

**i) Flexible Neighborhood Aggregation:** GPR filters employ learnable coefficients $\gamma$ to selectively aggregate neigh-

borhood information from different ranges. Unlike stacking static GNN layers, this approach prevents oversmoothing, preserving node-level distinctions while effectively capturing higher-order structural dependencies.

**ii) Diverse Structural Aspects:** As illustrated in Figure 1, each GPR filter captures unique structural neighboring aspects of nodes through learnable continuous coefficients, with their sign and magnitude enabling fine-grained emphasis or suppression of the corresponding neighborhoods. By leveraging multiple unique GPR filters with varying sizes, GPR-GAE can utilize diverse, multi-scale structural representations. This allows modeling of complex structures while mitigating adversarial smoothing by avoiding reliance on single representations derived from fixed aggregations.

We further extend our motivations in Appendix G, empirically showing how GPR-GAE moves beyond the simple proximity focus of traditional GAEs to capture complex structural properties. The appendix demonstrates GPR-GAE's ability to leverage higher-order information and distinguish between proximal connections, highlighting its advanced structural encoding capabilities and forming a foundation for its application in adversarial purification.

# 4. GPR-GAE

## 4.1. Node Encoding

We employ $K + 1$ distinct GPR filters to capture multi-scale structural information across varying neighborhood sizes. The output of the $k$-th GPR filter, $H_{\theta_k}$ ($k = 0, \ldots, K$), is:

$$\mathbf{H}_{\theta_k} = \sum_{m=0}^{k} \gamma_{k,m}(\tilde{\mathbf{A}}_{ns}^m \mathbf{H}^{(0)}), \quad \mathbf{H}^{(0)} = \mathbf{X} \cdot \mathbf{W}_n. \quad (4)$$

Here, $\gamma_k = [\gamma_{k,0}, \gamma_{k,1}, \ldots, \gamma_{k,k}]$ are the learnable coefficients for the $k$-th GPR filter, where $\gamma_{k,m}$ weighs the contribution of the $m$-hop neighborhood. $\mathbf{H}^{(0)} \in \mathbb{R}^{N \times Z_1}$ denotes the initial linearly transformed node embeddings, where $Z_1$ is the embedding dimension. Note that $\mathbf{H}_{\theta_0} = \mathbf{H}^{(0)}$ is fixed as the initial node embeddings. We use a normalized adjacency matrix without self-loops ($\tilde{\mathbf{A}}_{ns}$) to reduce potential excessive self-influence, promoting aggregation from distinct neighborhoods for more discriminative representations.

The final encoded node embedding, denoted as $\mathbf{H}^{\text{GPRGAE}} \in \mathbb{R}^{N \times (K+1) \cdot Z_1}$, is formed by concatenating the representations from all K+1 GPR filters:

$$\mathbf{H}^{\text{GPRGAE}} = \Big|\Big|_{k=0}^{K} \mathbf{H}_{\theta_k}. \quad (5)$$

This concatenation creates a rich, multi-scale representation for downstream tasks to effectively utilize.

## 4.2. Edge Encoding

The edge representation $\mathbf{E}_{i,j} \in \mathbb{R}^{Z_2}$, where $Z_2$ is the edge embedding dimension, is computed as:

$$\mathbf{E}_{i,j} = \phi(\mathbf{H}_i^{\text{GPRGAE}} || \mathbf{H}_j^{\text{GPRGAE}}) \cdot \mathbf{W}_e, \quad (6)$$

where the concatenation of encoded node embeddings of $i$ and $j$ are transformed by an activation function $\phi$ and a multiplication by a learnable weight matrix $\mathbf{W}_e$.

## 4.3. Edge Decoding

The encoded edge embeddings are order-variant, where $\mathbf{E}_{i,j}$ and $\mathbf{E}_{j,i}$ are two different representations. So we first compute the directed link prediction score:

$$\hat{\mathbf{A}}_{i \to j} = \sigma(\phi(\mathbf{E}_{i,j}) \cdot \mathbf{W}_d), \quad (7)$$

where $\mathbf{W}_d \in \mathbb{R}^{Z_2 \times 1}$ is a learnable weight vector and $\sigma$ is the sigmoid function. Given that our work focuses on undirected graphs, the final link prediction score $\hat{\mathbf{A}}_{ij}$ is obtained by averaging the bidirectional predictions:

$$\hat{\mathbf{A}}_{ij} = \frac{\hat{\mathbf{A}}_{i \to j} + \hat{\mathbf{A}}_{j \to i}}{2}. \quad (8)$$

The resulting $\hat{\mathbf{A}}_{ij}$ values form the predicted adjacency matrix $\hat{\mathbf{A}}$, where each entry represents the probability of an edge existing between the corresponding nodes.

# 5. Adversarial Purification

## 5.1. Perturbed Graph Sampling for Training

To sample perturbed graphs for training, we define the sample space $\mathcal{B}(G, (p, q, \eta))$, where we apply three random perturbation methods sequentially to $G$. The parameters $p, q, \eta$ represent the budgets for each perturbation method.

**i) Edge Injection.** We begin by injecting new edges into the graph $G = (\mathcal{V}, \mathcal{E})$. Specifically, we uniformly sample a set of edges $\mathcal{E}_{\text{inject}} \subseteq \mathcal{E}^C$, where $\mathcal{E}^C$ denotes the set of edges not present in $\mathcal{E}$. The number of injected edges is determined by the injection ratio $p$, such that $|\mathcal{E}_{\text{inject}}| = p \cdot |\mathcal{E}|$. These edges are added to the graph, resulting in edge set $\mathcal{E}' = \mathcal{E} \cup \mathcal{E}_{\text{inject}}$.

**ii) Edge Masking.** Next, we mask a subset of edges from the modified graph. The masked edge set is defined as $\mathcal{E}_{\text{mask}} = \mathcal{E}_{\text{mask\_orig}} \cup \mathcal{E}_{\text{mask\_inj}}$, where:

- $\mathcal{E}_{\text{mask\_orig}} \subseteq \mathcal{E}$ consists of original edges uniformly sampled from $\mathcal{E}$, with $|\mathcal{E}_{\text{mask\_orig}}| = q \cdot |\mathcal{E}|$,

- $\mathcal{E}_{\text{mask\_inj}} \subseteq \mathcal{E}_{\text{inject}}$ consists of injected edges uniformly sampled from $\mathcal{E}_{\text{inject}}$, with $|\mathcal{E}_{\text{mask\_inj}}| = q \cdot |\mathcal{E}_{\text{inject}}|$.

Edge masking results in an edge set $\mathcal{E}' = \mathcal{E}' \setminus \mathcal{E}_{\text{mask}}$.

**iii) Edge Reweighting.** Finally, we reweight the remaining edges in $\mathcal{E}'$. For each edge in $\mathcal{E}'$, originally weighted as 1, we assign a new weight randomly sampled from the range $[\beta, \beta \cdot \eta]$, where $\eta$ is a scaling factor. Note that the specific value of $\beta$ is inconsequential, as it is normalized during adjacency matrix processing.

**Resulting Perturbed Graph.** The perturbed graph sampled from $\mathcal{B}(G, (p, q, \eta))$ is defined as $G' = (\mathcal{V}, \mathcal{E}')$, where $\mathcal{E}'$ incorporates the effects of edge injection, masking, and reweighting. Equivalently, $G'$ can be represented as $(\mathbf{A}', \mathbf{X})$, where $\mathbf{A}'$ is a modified adjacency matrix with the continuous edge weights. While edge injection and edge masking define discrete boundaries for the sample space, edge reweighting smooths these boundaries into continuous space. This enables the learning of more generalized $\Delta G$, or more specifically, $\Delta \mathbf{A} = \mathbf{A} - \mathbf{A}'$, providing directions for purifying the graph structure toward its clean state.

### 5.2. Learning Objective

The learning objective of GPR-GAE is to purify the perturbed graph $G'$ by restoring its adjacency matrix $\mathbf{A}'$ to approximate the original $\mathbf{A}$. The model takes $G'$ as input and outputs link score predictions $\hat{\mathbf{A}}$, representing edge likelihoods in $\mathcal{E}_{all} = \mathcal{E} \cup \mathcal{E}_{inject}$. These predictions are optimized with the following two losses:

**(i) Restoration Loss.** The restoration loss quantifies the model's ability to recover the structure of the original graph. It optimizes the link prediction scores $\hat{\mathbf{A}}$ for positive edges in $\mathcal{E}$ and negative edges in $\mathcal{E}_{inject}$. The loss is defined as:

$$
\begin{aligned}
\mathcal{L}_{restore} = &\frac{1}{|\mathcal{E}|} \sum_{\{i,j\} \in \mathcal{E}} \log(1 - \hat{\mathbf{A}}_{ij}) \\
&+ \frac{1}{|\mathcal{E}_{inject}|} \sum_{\{i,j\} \in \mathcal{E}_{inject}} \log(\hat{\mathbf{A}}_{ij}).
\end{aligned} \quad (9)
$$

**(ii) Symmetry Loss.** The symmetry loss ensures consistency in link prediction scores for edges in $\mathcal{E}_{all}$ regardless of node ordering. It is defined as:

$$
\mathcal{L}_{sym} = \frac{1}{|\mathcal{E}_{all}|} \sum_{\{i,j\} \in \mathcal{E}_{all}} \left( \hat{\mathbf{A}}_{i \to j} - \hat{\mathbf{A}}_{j \to i} \right)^2. \quad (10)
$$

The total loss function combines these objectives:

$$
\mathcal{L}(\theta) = \mathcal{L}_{restore} + \delta \cdot \mathcal{L}_{sym}, \quad (11)
$$

where $\delta > 0$ balances restoration and symmetry losses, ensuring consistency throughout the predictions.

### 5.3. Multi-step Purification Process

During the test stage, the purification process is applied to the test graph $G_{test} = (\mathbf{A}_{test}, \mathbf{X}_{test})$. At each step $t$, the edge

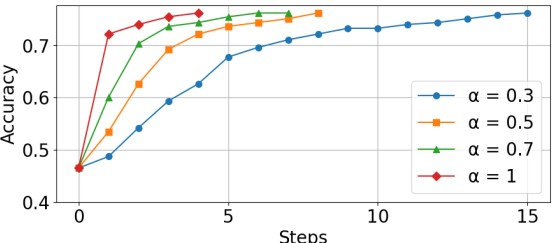

*Figure 2.* GCN classifier performance on attacked Cora. Node classification accuracy over purification steps using GPR-GAE multi-step purification with $\tau = 1/1000$, and $\alpha \in \{0.3, 0.5, 0.7, 1\}$.

weights in $\mathbf{A}_{test}$ are iteratively updated as:

$$
\begin{aligned}
\mathbf{A}^{(t+1)} &= \mathbf{A}^{(t)} + \alpha \cdot \Delta \mathbf{A}^{(t)}, \\
\Delta \mathbf{A}^{(t)} &= \hat{\mathbf{A}}^{(t)} - \mathbf{A}^{(t)}, \quad \hat{\mathbf{A}}^{(t)} = f_\theta(\mathbf{A}^{(t)}, \mathbf{X}_{test}).
\end{aligned} \quad (12)
$$

Here, $\mathbf{A}^{(0)} = \mathbf{A}_{test}$, and $\Delta \mathbf{A}^{(t)}$ adjusts the graph structure based on $\hat{\mathbf{A}}^{(t)}$. The step size $\alpha \in (0, 1]$ balances retaining the current state and incorporating updates. The process terminates when $\frac{\|\Delta \mathbf{A}^{(t)}\|}{\|\mathbf{A}^{(t)}\|} \leq \tau$. The final refined adjacency matrix, $\mathbf{A}^*$, is passed to the GNN classifier $f_\psi$ without discretization, producing the final classification $f_\psi(\mathbf{A}^*, \mathbf{X}_{test})$.

Figure 2 illustrates the accuracy across purification steps using GPR-GAE as the purifier. The classification accuracy improves with each step, demonstrating the effectiveness of the multi-step purification process. The results also show consistent convergence across all $\alpha$ values under the terminal condition. For practical use, we set $\alpha = 1$, $\tau = 1/1000$, and limit the maximum purification steps to 5.

### 5.4. Computational Complexity

In real-world graphs, where the number of nodes is $N$, the number of edges is $|\mathcal{E}|$, the feature and the hidden dimension is $Z$, the complexity of node encoding in GPR-GAE is $\mathcal{O}(K \cdot |\mathcal{E}| \cdot Z)$, the same as GCN with $K$-layers. For multi-step purification process with the number of maximum purification steps fixed as 5, the complexity is $\mathcal{O}(K \cdot |\mathcal{E}| \cdot Z^2)$. We provide detailed derivations in Appendix B.

## 6. Experiments

We conducted experiments on various datasets including Cora, Cora_ML, Citeseer (Bojchevski & Günnemann 2018), Pubmed (Sen et al. 2008), OGB-arXiv (Hu et al. 2020), and Chameleon with removed duplicates (Platonov et al. 2023). We use an inductive split with 20 labeled nodes per class for train and validation, a stratified test set of 10% of nodes, and the remaining nodes as unlabeled training data. For Chameleon and OGB-arXiv, we use their provided splits with fully labeled training sets.

*Table 1.* **Adaptive attack:** Test accuracy (%) on Cora for models with different methods: Vanilla (standard training), Self-training (pseudo-labeling), Adversarial training (PRBCD or LRBCD with training $\epsilon = 0.2$), and GPR-GAE$_{\text{GNN}}$ paired with either vanilla or self-trained classifiers. Robust GNNs (EvenNet, SoftMedianGDC) are compared alongside GCN variants. Results are evaluated under clean conditions and adversarial attacks (LRBCD and PRBCD) with varying budget $\epsilon$. Bold values indicate the best performance within each category, while curly underlined values highlight the second-best performance.

| Model | Adv. Pur. | Pseu. Lab. | (Adv.eval) → (Adv.tr) ↓ | Clean | *LRBCD* $\epsilon = 0.1$ | *PRBCD* $\epsilon = 0.1$ | *LRBCD* $\epsilon = 0.25$ | *PRBCD* $\epsilon = 0.25$ | *LRBCD* $\epsilon = 0.5$ | *PRBCD* $\epsilon = 0.5$ |
|---|---|---|---|---|---|---|---|---|---|---|
| EvenNet | ✗ | ✗ | ✗ | 81.4±2.0 | 65.2±2.3 | 65.7±1.4 | 54.9±1.1 | 51.7±3.3 | 45.7±0.5 | 35.6±0.9 |
| SoftMedianGDC | ✗ | ✗ | ✗ | 77.4±1.8 | 69.1±1.9 | 67.3±1.9 | 64.1±1.4 | 60.0±1.4 | 59.1±1.2 | 48.2±1.1 |
| GCN Vanilla | ✗ | ✗ | ✗ | 79.4±0.7 | 64.2±2.3 | 60.1±1.0 | 51.2±0.6 | 46.9±1.5 | 40.5±2.2 | 29.3±2.3 |
| GCN Self-trained | ✗ | ✓ | ✗ | 82.5±0.9 | 70.7±1.6 | 65.0±0.9 | 60.7±1.7 | 52.7±1.2 | 48.4±1.1 | 37.2±2.1 |
| GCN Adv.-trained | ✗ | ✓ | *LRBCD* | 80.5±1.8 | 70.4±3.2 | 63.8±1.5 | 62.8±2.7 | 51.1±1.4 | 52.6±2.8 | 33.4±1.6 |
| GCN Adv.-trained | ✗ | ✓ | *PRBCD* | 80.3±1.7 | 69.5±1.5 | 64.0±1.7 | 61.6±2.0 | 51.7±2.4 | 51.7±1.8 | 34.6±1.9 |
| **GPR-GAE$_{\text{GCN\_Vanilla}}$** | ✓ | ✗ | ✗ | 79.4±0.3 | 75.9±2.0 | 75.6±2.1 | 74.1±1.4 | **69.7±2.9** | 72.5±1.7 | **61.3±3.6** |
| **GPR-GAE$_{\text{GCN\_Self}}$** | ✓ | ✓ | ✗ | 81.6±0.7 | **78.5±1.6** | **77.3±0.9** | **76.6±1.8** | 69.0±1.2 | **75.0±1.6** | 60.8±1.4 |
| GAT Vanilla | ✗ | ✗ | ✗ | 76.4±1.3 | 50.2±2.4 | 50.5±4.0 | 29.5±3.1 | 36.2±4.2 | 19.8±2.4 | 21.4±1.5 |
| GAT Self-trained | ✗ | ✓ | ✗ | 79.4±2.3 | 53.5±6.1 | 53.0±4.6 | 34.9±6.1 | 36.5±5.8 | 24.1±6.7 | 24.8±5.0 |
| GAT Adv.-trained | ✗ | ✓ | *LRBCD* | 80.7±0.8 | 67.4±0.7 | 62.0±0.8 | 59.4±2.4 | 48.4±0.5 | 50.5±2.7 | 32.2±1.3 |
| GAT Adv.-trained | ✗ | ✓ | *PRBCD* | 78.8±1.7 | 65.2±4.2 | 62.1±3.5 | 50.6±3.8 | 51.2±4.6 | 38.1±6.9 | 39.0±7.3 |
| **GPR-GAE$_{\text{GAT\_Vanilla}}$** | ✓ | ✗ | ✗ | 76.0±1.6 | 71.3±2.1 | 71.8±2.4 | 69.0±1.6 | 65.1±1.1 | 67.1±1.3 | 59.9±3.0 |
| **GPR-GAE$_{\text{GAT\_Self}}$** | ✓ | ✓ | ✗ | 79.4±2.2 | **74.9±2.8** | **74.0±3.6** | **72.8±3.9** | **69.3±5.4** | **71.7±4.5** | **60.5±5.3** |
| GPRGNN Vanilla | ✗ | ✗ | ✗ | 81.4±1.2 | 65.6±0.7 | 63.0±1.4 | 55.6±0.9 | 49.4±1.9 | 44.2±2.7 | 33.1±3.2 |
| GPRGNN Self-trained | ✗ | ✓ | ✗ | 81.1±1.8 | 68.5±1.3 | 65.5±2.2 | 60.2±2.3 | 51.8±1.3 | 51.9±2.0 | 36.8±1.9 |
| GPRGNN Adv.-trained | ✗ | ✓ | *LRBCD* | 80.0±1.0 | 71.2±3.0 | 65.4±1.1 | 64.8±2.6 | 54.0±1.2 | 57.7±4.4 | 38.2±2.7 |
| GPRGNN Adv.-trained | ✗ | ✓ | *PRBCD* | 79.5±2.0 | 73.2±2.8 | 68.0±2.4 | 68.5±3.1 | 61.1±4.4 | 62.8±5.6 | 48.5±5.3 |
| **GPR-GAE$_{\text{GPRGNN\_Vanilla}}$** | ✓ | ✗ | ✗ | 81.6±1.1 | **77.4±1.4** | **77.0±0.8** | **76.4±0.9** | 70.6±2.3 | **75.6±1.9** | **64.2±1.0** |
| **GPR-GAE$_{\text{GPRGNN\_Self}}$** | ✓ | ✓ | ✗ | 80.7±1.6 | 77.2±1.0 | 75.9±1.2 | 75.5±1.1 | 70.2±1.6 | 74.7±1.3 | 61.6±1.0 |
| APPNP Vanilla | ✗ | ✗ | ✗ | 82.1±1.2 | 66.2±1.3 | 64.5±0.8 | 56.4±1.3 | 50.9±1.2 | 46.0±2.5 | 32.6±2.6 |
| APPNP Self-trained | ✗ | ✓ | ✗ | 82.3±1.8 | 70.6±1.0 | 68.2±1.1 | 62.1±2.8 | 54.7±1.0 | 51.1±3.2 | 37.7±2.0 |
| APPNP Adv.-trained | ✗ | ✓ | *LRBCD* | 82.4±1.5 | 71.5±1.3 | 66.8±0.6 | 63.4±2.8 | 54.0±1.4 | 54.4±1.8 | 35.6±2.3 |
| APPNP Adv.-trained | ✗ | ✓ | *PRBCD* | 81.2±1.5 | 71.2±1.4 | 67.6±0.6 | 64.1±1.3 | 55.3±1.3 | 54.1±0.7 | 37.8±3.2 |
| **GPR-GAE$_{\text{APPNP\_Vanilla}}$** | ✓ | ✗ | ✗ | 81.6±1.4 | 77.6±1.8 | 77.1±1.5 | 77.3±1.7 | **72.4±1.9** | 76.4±1.8 | 64.3±2.7 |
| **GPR-GAE$_{\text{APPNP\_Self}}$** | ✓ | ✓ | ✗ | **82.5±1.7** | **79.6±2.3** | **77.3±2.9** | **77.9±2.2** | 71.4±2.3 | **76.5±2.7** | **65.2±2.0** |

**GPR-GAE.** We set $K = 7$, symmetry loss factor $\delta = 0.2$, and the training sample budget to $(p, q, \eta) = (1.5, 0.2, 3)$, with $q = 0.5$ tuned for OGB-arXiv. We provide ablation studies of GPR-GAE in Appendix F. Additionally, GPR-GAE$_{\text{GNN}}$ denotes a GNN classifier performing inference on the purified graph produced by GPR-GAE.

**Attacks.** PRBCD (Geisler et al. 2021) is a gradient-based topology attack that perturbs graph structures by optimizing an attack objective over randomized edge blocks, adhering to a defined perturbation budget. Its local constraint variant, LRBCD (Gosch et al. 2023), applies restrictions to perturbations within a node's local neighborhood, simulating more realistic attack scenarios. The $\epsilon$ for attacks parametrizes the global budget $\Delta = \left| \epsilon \cdot \sum_{v \in \mathcal{V}_t} d_v / 2 \right|$ relative to the degree $d_v$ for the set of targeted nodes $\mathcal{V}_t$.

**Adversarial Training.** For adversarial training, we follow the pseudo-labeling framework proposed by Gosch et al. (2023). For datasets with unlabeled training nodes, a separate classifier is pre-trained on labeled nodes to assign previously unlabeled nodes with pseudo-labels, expanding the labeled training set. The final model is then trained on the expanded labeled training set, using either PRBCD or LRBCD to create adversarial training samples. Additionally the non-adversarial counterpart, self-training, uses pseudo-labeling without adversarial training, and the expansion of the training set typically increases the robustness of both adversarial and self-training through generalization.

### 6.1. Experimental Results

**Adaptive attack.** We evaluate robustness under adaptive attacks, where the attacker has full knowledge of the model's architecture and parameters, crafting model-specific worst-case perturbations. Our adversarial purification method, GPR-GAE, is tested in two configurations—attached to either vanilla or self-trained classifiers—and compared against vanilla, self-training, adversarial training, and robust GNNs (EvenNet, SoftMedianGDC). For the baseline classifier, we employ four GNN models: GCN, GAT (Veličković et al. 2018), GPRGNN, and APPNP (Gasteiger et al. 2019). Note that GPRGNN demonstrated state-of-the-art robustness under adversarial training in the prior work, flexibly learning robust propagation pathways in adversarial settings.

Table 1 highlights that, on the Cora dataset, GPR-GAE achieves significantly higher robustness compared to all

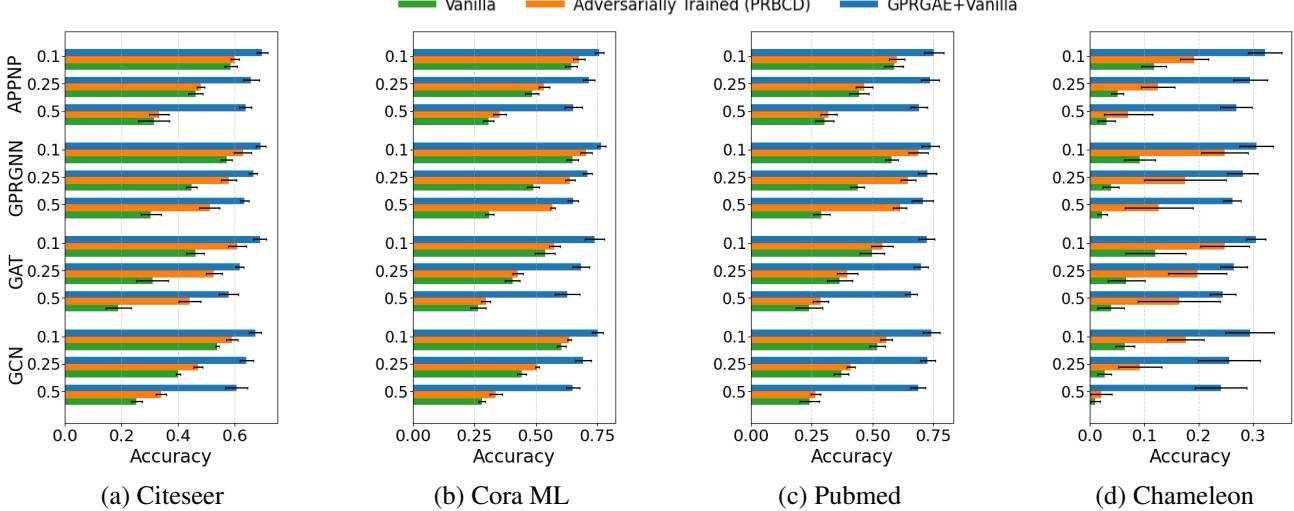

*Figure 3.* **Adaptive Attack:** Comparison of test accuracy (%) for Vanilla, Adversarial Training (PRBCD with $\epsilon = 0.2$), and GPR-GAE$_{\text{GNN\_Vanilla}}$ under PRBCD attacks with perturbation budgets $\epsilon = 0.1, 0.25, 0.5$ on various datasets and GNN classifiers.

other GNN model variants, including robust GNNs, under adaptive attacks. For example, while adversarially trained GPRGNN (PRBCD)—the most robust method aside from GPR-GAE—achieves 48.5% test accuracy against PRBCD attacks with $\epsilon = 0.5$, GPR-GAE$_{\text{GPRGNN\_Vanilla}}$ improves this by 15.7 percentage points through its multi-step refinements, demonstrating its robustness under severe perturbations. Against LRBCD attacks, which restrict local edge modifications proportionally to node degrees, GPR-GAE maintains strong performance, with its largest accuracy drop relative to clean data being only 8.9 percentage points for GPR-GAE$_{\text{GAT\_Vanilla}}$. This underscores GPR-GAE's significance in defending against more realistic attack scenarios.

In terms of clean accuracy, adversarially trained (PRBCD) GPRGNN sacrifices 1.6 percentage points compared to its non-adversarial counterpart, self-trained GPRGNN. In contrast, GPR-GAE shows minimal trade-offs, with the largest drop being 0.9 points when paired with self-trained GCN. These results demonstrate that GPR-GAE successfully mitigates the trade-off by decoupling accuracy and robustness into separate modules with different specialties, achieving strong robustness against adaptive attacks while preserving clean accuracy. Notably, for GPR-GAE, overall performance against attacks tends to correlate with the initial clean accuracy of the base classifier, rather than the type of GNN or whether it was self-trained or not.

In Figure 3, we further compare the robustness of GPR-GAE$_{\text{GNN\_Vanilla}}$ against Vanilla and Adversarial Training (PRBCD) across various datasets, including three homophilic datasets (Citeseer, Cora ML, Pubmed) and a heterophilic dataset (Chameleon). The results show that GPR-

*Table 2.* **Non-adaptive attack:** Average test accuracy (%) in adversarial purification methods and robust GNNs under clean conditions and transfer attacks (PRBCD) from vanilla GCN. "OOM" indicates "Out-Of-Memory."

| Model | Citeseer (small) Clean / 0.25 / 0.5 | Pubmed (medium) Clean / 0.25 / 0.5 | OGB-arXiv (large) Clean / 0.25 / 0.5 |
|---|---|---|---|
| *GCN (Vanilla)* | *71.5 / 40.0 / 25.3* | *75.1 / 37.1 / 24.1* | *70.8 / 43.1 / 33.1* |
| EvenNet | **72.9** / 57.0 / 48.0 | 75.0 / 64.5 / 57.3 | 66.2 / 46.7 / 39.0 |
| SoftMedianGDC | 71.7 / 52.1 / 37.5 | 73.9 / 53.5 / 40.2 | **70.5** / 44.4 / 35.7 |
| Jaccard-GCN | 69.8 / 50.4 / 40.6 | 74.3 / 46.4 / 36.0 | 67.6 / 43.9 / 35.0 |
| SVD-GCN | 66.7 / 59.8 / 46.7 | 70.7 / 68.0 / 61.4 | OOM |
| GOOD-AT | 68.1 / 60.8 / 56.6 | 73.0 / 70.5 / 68.9 | OOM |
| **GPR-GAE$_{\text{GCN\_Vanilla}}$** | 71.2 / **68.7** / **66.4** | **75.1 / 73.5 / 72.0** | 69.1 / **49.2** / **42.3** |

GAE consistently achieves superior defense performance across all four GNN models and datasets, demonstrating both its effectiveness and broad applicability.

**Non-adaptive attack.** Unlike GPR-GAE, prior adversarial purification methods such as SVD-GCN, Jaccard-GCN, and GOOD-AT rely on discrete, non-differentiable purification, preventing direct gradient-based adaptive attacks and each requiring different attack strategies. For general evaluation of adversarial purification methods, we conduct experiments under non-adaptive settings, transferring attacks from adaptively attacked vanilla GCNs using PRBCD. In Table 2, we compare GPR-GAE$_{\text{GCN\_Vanilla}}$ with prior adversarial purification methods and robust GNNs across datasets of varying scales: Citeseer ($N = 2,110$, $|\mathcal{E}| = 3,668$), Pubmed ($N = 19,717$, $|\mathcal{E}| = 44,324$), and OGB-arXiv ($N = 169,343$, $|\mathcal{E}| = 1,157,799$).

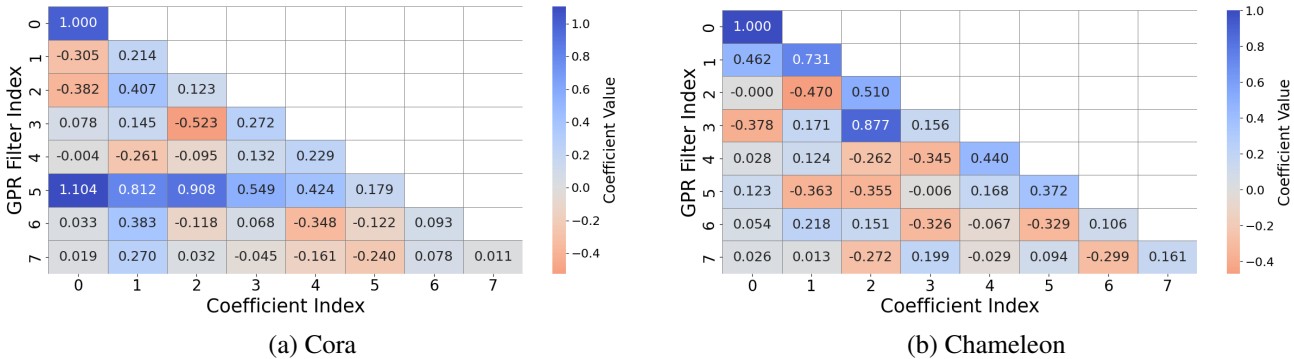

(a) Cora                                    (b) Chameleon

*Figure 4.* Visualization of the learned coefficients for each GPR filter in GPR-GAE. For the coefficient value $\gamma_{i,j}$, $i$ indicates the GPR Filter Index ($i$-th GPR Filter) and $j$ indicates the Coefficient Index (for $j$-th hop). We adjust the sign of the values so that the last coefficient values of each GPR filter are positive.

*Table 3.* Empirical Lipschitz constant $\hat{L}$.

| Cora | Cora ML | Citeseer | Pubmed | OGB-arXiv | Chameleon |
|------|---------|----------|--------|-----------|-----------|
| 0.3988 | 0.4587 | 0.2788 | 0.4473 | 0.4214 | 0.3238 |

Heuristic-based approaches such as Jaccard-GCN and SVD-GCN exhibit either a significant drop in clean accuracy or limited robustness, making them less adaptable across datasets. GOOD-AT, which detects adversarial edges using classifier-based embeddings, provides improved robustness over heuristics but does not fully utilize structural information, falling short in overall compared to GPR-GAE. Moreover, training of GOOD-AT relies on iterative dense matrix optimization for adversarial sample generation, making it impractical for large graphs like OGB-arXiv, a limitation also present in SVD-GCN's low-rank approximation.

While robust GNNs (EvenNet, SoftMedianGDC) may achieve higher clean accuracy, GPR-GAE maintains superior adversarial robustness, generally with larger margins under stronger perturbations, while preserving clean accuracy close to its attached classifier, vanilla GCN. Additionally, it scales efficiently to large datasets like OGB-arXiv by leveraging mini-batch processing, training with 1% of edges per epoch, and purifying the test stage graphs in batches—ensuring both memory efficiency and robustness.

### 6.2. Empirical Analysis

**Empirical estimation of Lipschitz constant.** To support Theorem 3.3, we empirically estimate the local Lipschitz constant of the model on the test stage graph, where the output after a single step purification is denoted as $f_\theta(\mathbf{A}, \mathbf{X})$. We randomly sample 100 pairs of perturbed adjacency matrices $(\mathbf{A}_1, \mathbf{A}_2)$, each generated by injecting random edges into the test graph with a random perturbation budget under

0.5. For each pair, we compute:

$$\frac{\|f_\theta(\mathbf{A}_1, \mathbf{X}_{\text{test}}) - f_\theta(\mathbf{A}_2, \mathbf{X}_{\text{test}})\|_F}{\|\mathbf{A}_1 - \mathbf{A}_2\|_F},$$

and report the maximum value across the 100 sampled pairs as the empirical Lipschitz constant $\hat{L}$. As shown in Table 3, all values remain strictly below 1, supporting the convergent nature of our multi-step purification framework.

**Unique learned filters.** In Figure 4, we visualize the learned coefficients of GPR-GAE under a homophilic graph (Cora) and a heterophilic graph (Chameleon). Under supervised settings, as discussed in Chien et al. (2021), GPRGNN tends to learn a single GPR filter, with coefficients exhibiting consistent signs on homophilic graphs and fluctuating patterns on heterophilic ones. In contrast, GPR-GAE, trained in our self-supervised framework, learns diverse coefficients regardless of the graph's homophily. This diversity indicates that each GPR filter selectively includes or suppresses neighborhood information, resulting in distinct multi-scale structural representations. We provide visualization for other datasets in Appendix D, where the variation in learned coefficients across datasets further highlights GPR-GAE's adaptability as a data-driven purification model.

## 7. Conclusion

We propose a self-supervised adversarial purification framework for GNNs, explicitly decoupling accuracy and robustness to mitigate their trade-off. At its core is GPR-GAE, a specialized purifier trained in a self-supervised manner, leveraging multi-scale neighborhood information for enhanced structural learning. Through multi-step purification, GPR-GAE demonstrates state-of-the-art robustness across diverse datasets and attacks, particularly under high perturbations, while preserving clean accuracy, making it a versatile plug-and-play defense module for GNN classifiers.

## Impact Statement

This paper presents work whose goal is to advance the field of Machine Learning. While our work may have potential societal implications, none require explicit highlighting in this context.

## Acknowledgements

This work was supported by the Institute of Information & Communications Technology Planning & evaluation (IITP) grant and the National Research Foundation of Korea (NRF) grant funded by the Korean government (MSIT) (RS-2019-II190421, IITP-2025-RS-2020-II201821, RS-2024-00438686, RS-2024-00436936, RS-2023-00225441, RS-2024-00448809, IITP-2025-RS-2024-00360227, RS-2025-02218768, RS-2025-25443718, RS-2025-25442569, RS-2025-02653113).

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

# A. Theoretical Proofs

## A.1. Proof of Theorem 3.1: Decomposition of Adversarial Training

We prove the decomposition of the adversarial training objective into accuracy and robustness terms.

*Proof.* The learning objective of adversarial training is:

$$\mathcal{L}(\psi) = \max_{G' \in \mathcal{B}(G,\epsilon)} \mathbb{E}_{v \in \mathcal{V}} \big[ \ell(f_\psi(G', v), y_v) \big] \tag{1}$$

$$= \max_{G' \in \mathcal{B}(G,\epsilon)} \frac{1}{|\mathcal{V}|} \sum_{v \in \mathcal{V}} \ell(f_\psi(G', v), y_v). \tag{2}$$

We partition $\mathcal{V}$ into disjoint sets $\mathcal{V}_{\text{unaffected}}$ and $\mathcal{V}_{\text{affected}}$, representing nodes unaffected and affected by perturbations, respectively. Then:

$$\sum_{v \in \mathcal{V}} \ell(f_\psi(G', v), y_v) = \sum_{v \in \mathcal{V}_{\text{unaffected}}} \ell(f_\psi(G', v), y_v) + \sum_{v \in \mathcal{V}_{\text{affected}}} \ell(f_\psi(G', v), y_v). \tag{3}$$

For all $\mathcal{V}_{\text{unaffected}}$, the predictions remain unchanged under perturbation, and thus the corresponding losses are identical:

$$\sum_{v \in \mathcal{V}_{\text{unaffected}}} \ell(f_\psi(G', v), y_v) = \sum_{v \in \mathcal{V}_{\text{unaffected}}} \ell(f_\psi(G, v), y_v). \tag{4}$$

For $\mathcal{V}_{\text{affected}}$, adversarial perturbations maximize the loss:

$$\sum_{v \in \mathcal{V}_{\text{affected}}} \ell(f_\psi(G', v), y_v) = \max_{G' \in \mathcal{B}(G,\epsilon)} \sum_{v \in \mathcal{V}_{\text{affected}}} \ell(f_\psi(G', v), y_v). \tag{5}$$

Substituting back:

$$\mathcal{L}(\psi) = \frac{|\mathcal{V}_{\text{unaffected}}|}{|\mathcal{V}|} \cdot \frac{1}{|\mathcal{V}_{\text{unaffected}}|} \sum_{v \in \mathcal{V}_{\text{unaffected}}} \ell(f_\psi(G, v), y_v) + \frac{|\mathcal{V}_{\text{affected}}|}{|\mathcal{V}|} \cdot \frac{1}{|\mathcal{V}_{\text{affected}}|} \max_{G' \in \mathcal{B}(G,\epsilon)} \sum_{v \in \mathcal{V}_{\text{affected}}} \ell(f_\psi(G', v), y_v). \tag{6}$$

Let $\lambda = \frac{|\mathcal{V}_{\text{unaffected}}|}{|\mathcal{V}|}$. Finally, rewriting as expectations:

$$\mathcal{L}(\psi) = \lambda \cdot \mathbb{E}_{v \in \mathcal{V}_{\text{unaffected}}} \big[ \ell(f_\psi(G, v), y_v) \big] + (1 - \lambda) \cdot \max_{G' \in \mathcal{B}(G,\epsilon)} \mathbb{E}_{v \in \mathcal{V}_{\text{affected}}} \big[ \ell(f_\psi(G', v), y_v) \big]. \tag{7}$$

$\square$

## A.2. Proof of Theorem 3.3: Convergence of Multi-Step Purification

*Proof.* Building on the locally Lipschitz continuity of $f_\theta$ around $G^{(0)}$ with constant $L \in [0, 1)$, we consider two cases based on the choice of $\alpha$.

When $\alpha = 1$, the update reduces to direct application of $f_\theta$:

$$G^{(t+1)} = f_\theta(G^{(t)}). \tag{8}$$

Since $f_\theta$ is locally Lipschitz continuous with constant $L \in [0, 1)$, it is a contraction mapping.

When $0 < \alpha < 1$, we can define the update operator $h_\theta$ as:

$$h_\theta(G) = (1 - \alpha)G + \alpha f_\theta(G). \tag{9}$$

For any $G_1, G_2$ in the neighborhood, we have

$$
\begin{aligned}
\|h_\theta(G_1) - h_\theta(G_2)\| &= \|(1-\alpha)(G_1 - G_2) + \alpha(f_\theta(G_1) - f_\theta(G_2))\| \\
&\leq \|(1-\alpha)(G_1 - G_2)\| + \|\alpha(f_\theta(G_1) - f_\theta(G_2))\| \\
&\leq (1-\alpha)\|G_1 - G_2\| + \alpha L\|G_1 - G_2\| \\
&= (1 - \alpha(1-L))\|G_1 - G_2\|.
\end{aligned}
\tag{10}
$$

Since $L < 1$ and $\alpha > 0$, the contraction factor $1 - \alpha(1-L)$ lies in $[0, 1)$, so $h_\theta$ is also a contraction mapping.

In both cases, the purification update defines a contraction mapping. Thus, by Banach's Fixed-Point Theorem, the sequence $(G^{(t)})_{t \geq 0}$ converges linearly to a unique fixed point $G^*$.

$\square$

## B. Derivation of Computational Complexity

### B.1. Node Encoding in GPR-GAE

The computational complexity of node encoding in GPR-GAE is derived as follows:

- **Initial Node Representation:** Computing the initial node embedding $\mathbf{H}^{(0)}$ through linear transformation requires:

$$
\mathcal{O}(N \cdot Z^2).
$$

- **Hop Representations:** The $k$-th hop representation is computed iteratively using:

$$
\mathbf{H}^{(k)} = \tilde{\mathbf{A}}_{ns}\mathbf{H}^{(k-1)},
$$

where $\tilde{\mathbf{A}}_{ns}$ is the normalized adjacency matrix. Each sparse matrix multiplication costs $\mathcal{O}(|\mathcal{E}| \cdot Z)$. Since this process is repeated for all $K$ hops (from $\mathbf{H}^{(1)}$ to $\mathbf{H}^{(K)}$), the total cost for computing all hop representations is:

$$
\mathcal{O}(K \cdot |\mathcal{E}| \cdot Z).
$$

- **GPR Filter Representations:** For each of the $k$-th GPR filter, the representation $\mathbf{H}_{\theta_k}$ is computed as:

$$
\mathbf{H}_{\theta_k} = \sum_{m=0}^{k} \gamma_{k,m}\mathbf{H}^{(m)},
$$

where $\mathbf{H}^{(m)} \in \mathbb{R}^{N \times Z}$. Summing all $k$ hop representations for $k$-th GPR filter costs:

$$
\mathcal{O}(k \cdot N \cdot Z).
$$

Summing over all $K + 1$ filters, the total cost becomes:

$$
\mathcal{O}\left(\sum_{k=0}^{K} k \cdot N \cdot Z\right) \approx \mathcal{O}(K^2 \cdot N \cdot Z).
$$

**Total Node Encoding Complexity:** Combining all components:

$$
\mathcal{O}(N \cdot Z^2 + K \cdot |\mathcal{E}| \cdot Z + K^2 \cdot N \cdot Z).
$$

In real-world graphs, where $N \ll |\mathcal{E}|$, and given $K < 10$, the final node encoding complexity can be simplified to the dominant term:

$$
\mathcal{O}(K \cdot |\mathcal{E}| \cdot Z).
$$

## B.2. Multi-Step Purification Complexity

The multi-step purification process iteratively updates the graph structure and includes three key components:

- **Node Encoding:** Each purification step first encodes each node, which costs:

$$\mathcal{O}(K \cdot |\mathcal{E}| \cdot Z).$$

- **Edge Encoding:** Each encoded edge embedding with dimension $Z$ is computed by transforming the concatenation of two encoded node embeddings. The combined input dimension is $2(K+1) \cdot Z$, resulting in a complexity of:

$$\mathcal{O}(2(K+1) \cdot |\mathcal{E}| \cdot Z^2).$$

  Simplifying the constants give:

$$\mathcal{O}(K \cdot |\mathcal{E}| \cdot Z^2).$$

- **Edge Decoding:** The edge decoder transforms the edge embeddings from $Z$ dimensions back to a single scalar. This costs:

$$\mathcal{O}(|\mathcal{E}| \cdot Z).$$

**Complexity per Step:** Summing the costs of node encoding, edge encoding, and edge decoding, the overall complexity for a single purification step is:

$$\mathcal{O}(K \cdot |\mathcal{E}| \cdot Z + K \cdot |\mathcal{E}| \cdot Z^2 + |\mathcal{E}| \cdot Z).$$

The complexity simplifies to:

$$\mathcal{O}(K \cdot |\mathcal{E}| \cdot Z^2).$$

**Total Multi-step Purification Complexity:** Assuming a fixed number of purification steps $T$, the total complexity for multi-step purification becomes:

$$\mathcal{O}(T \cdot K \cdot |\mathcal{E}| \cdot Z^2).$$

With $T$ treated as a small constant of 5, this simplifies further to:

$$\mathcal{O}(K \cdot |\mathcal{E}| \cdot Z^2).$$

# C. Experimental Settings

*Table C.1.* Datasets

| Dataset | Nodes | Edges | Features | Classes | Homophily Rate |
|---------|-------|-------|----------|---------|----------------|
| Cora | 2,708 | 5,278 | 1,433 | 7 | 0.80 |
| Cora-ML | 2,810 | 7,981 | 2,879 | 7 | 0.78 |
| Citeseer | 2,110 | 3,668 | 3,703 | 6 | 0.73 |
| Chameleon | 890 | 8,854 | 2,325 | 5 | 0.23 |
| Pubmed | 19,717 | 44,324 | 500 | 3 | 0.80 |
| OGB-arXiv | 169,343 | 1,157,799 | 128 | 40 | 0.65 |

In this section, we summarize the experimental settings, including GPR-GAE, other models, attacks, and adversarial training. For adversarial training and models other than GPR-GAE, we mostly follow Gosch et al. (2023), the prior work we use as the baseline for adversarial training. All experiments are conducted on an NVIDIA A100 (80GB) GPU. However, it is worth noting that GPR-GAE can be trained and applied to datasets, including OGB-arXiv, using an NVIDIA RTX A5000 (24GB). From an attacker's perspective, however, adaptively attacking classifiers attached to GPR-GAE using gradient-based methods introduces significant memory constraints. This is because these attacks require storing all edge embeddings from GPR-GAE for gradient computation, including the blocks of edges sampled at each epoch in PRBCD attack.

While GPR-GAE can be applied to OGB-arXiv by purifying the graph in batches, the gradient-based attacks cannot leverage the batching strategy in their gradient computations for generating attacks. As a result, adaptive attacks on GPR-GAE become unscalable (which could be seen as an advantage on our part), and we evaluate robustness on OGB-arXiv only in transfer attack settings. Furthermore, PGD attacks (Xu et al., 2019a) are particularly unscalable, as they require computing gradients for all possible edges, making them impractical to attack GPR-GAE even on small-scale graphs like Citeseer.

## C.1. Models

- **GCN:** Two-layer GCN with 64 hidden units. For OGB-arXiv, a three-layer GCN with 256 hidden units. While training, a dropout of 0.5 is applied.

- **GAT:** Two-layer GAT with 64 hidden units. Single attention head and 0.5 dropout during training.

- **GPRGNN:** Two-layer MLP with 64 hidden units for the initial feature transformation. The GPR coefficients are randomly initialized while a total of $K = 10$ diffusion steps are applied with 0.2 MLP dropout during training.

- **APPNP:** Two-layer MLP with 64 hidden units for the initial feature transformation. A total of $K = 10$ diffusion steps are applied with 0.5 MLP dropout during training. For the coefficients, $\gamma_K = (1 - \alpha)^K$ and $\gamma_l = \alpha(1 - \alpha)^l$ for $l < K$, with $\alpha$ fixed as 0.1.

- **GPR-GAE:** We set $K = 7$, $Z_1 = 128$, and $Z_2 = 512$, using the ELU activation function. The GPR coefficients are initialized randomly. During training, we apply an MLP dropout rate of 0.7 for node encoding, while no dropout is used for edge encoding or decoding. For the OGB-arXiv dataset, we use mini-batch training with a batch ratio $\lambda = 0.01$, sampling 1% of the original and inserted edges for each training epoch. During the test stage, adjacency matrix predictions are performed in batches, which does not affect the performance purification process. Full-batch training and purification are applied to all other datasets.

- **Jaccard-GCN:** The edges are filtered based on Jaccard dissimilarity with a threshold of 0.01. For the model, we use the DeepRobust library (Li et al. 2021), with modifications for ogbn-arxiv, where we use a three-layer GCN with 256 hidden units and prune edges using cosine dissimilarity with a threshold 0.4, as the dataset has non-binary attributes.

- **SVD-GCN:** Adversarial perturbations are filtered by using a low-rank approximation. For the model, we use the DeepRobust library.

- **GOOD-AT:** We follow the settings in https://github.com/likuanppd/GOOD-AT. 20 MLP detectors with 64 hidden units are trained with a learning rate of 0.01 and weight decay of 0.0001 using the ADAM optimizer. Each detector is trained on different adversarial samples generated through attacking a GCN classifier. A threshold of 0.1 is used, where edges with scores that exceed the threshold are detected as Out-Of-Distribution and pruned.

- **EvenNet:** Two-layer MLP with 64 hidden units (256 for OGB-arXiv) for initial feature transformation. A total of $K = 10$ is used with PPR initialization $\alpha = 0.1$. We use a dropout rate of 0.5 during training.

- **SoftMedianGDC:** The default configurations of Geisler et al. (2021) are used. Two-layer GDC with 64 hidden units, temperature $T = 0.2$ for the SoftMedian aggregation, Personalized PageRank diffusion coefficent $\alpha = 0.15$, and $k = 64$ for sparsification. For OGB-arXiv, a three-layer GDC with 256 hidden units is used with $T = 5$ and $\alpha = 0.1$.

## C.2. Training settings

GPR-GAE is trained using the ADAM optimizer with a learning rate of 0.01 and a weight decay of 0.0001. Training is conducted for 2000 epochs. We create 10 validation edge sets by randomly inserting negative edges with $p = 0.3 \cdot i$. Here, $i$ indicates the $i$-th validation edge set. We evaluate the model every epoch by computing the mean AUC and mean AP across the 10 validation edge sets. The model with the best performance is selected based on these metrics. When training the classifiers, a maximum of 3000 epochs is used for training, using the Adam optimizer with a learning rate of 0.01, weight decay of 0.001, and tanhMargin loss. An early stop method is used with a patience of 200 epochs. For adversarial training, the first ten epochs are trained without adversarial examples.

For attack parameters in adversarial training, we use 20 epochs with no early stopping. The block size is 1 million, and the loss type is tanhMargin. For PRBCD, the learning factor is 100, while for LRBCD, the learning factor is 20 times that of PRBCD. When training, a budget of $\epsilon = 0.2$ is used.

## C.3. Choice of Hyperparameters of Attacks for Evaluation

For attack parameters in the evaluation stage, we use a learning factor of 100 with 400 epochs. In the case of PRBCD, 100 additional epochs are performed with a decaying learning rate and without block resampling.

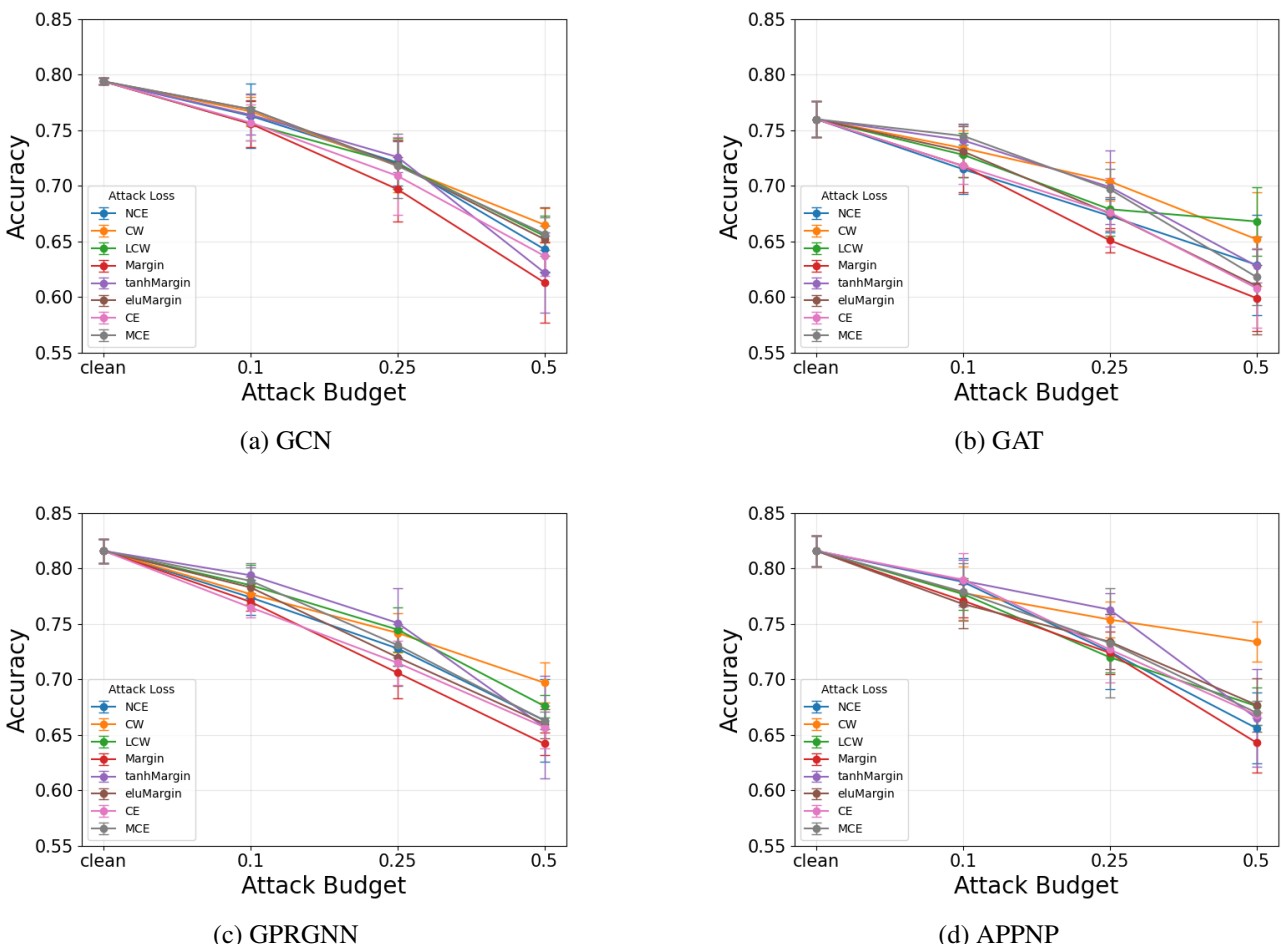

(a) GCN            (b) GAT

(c) GPRGNN            (d) APPNP

*Figure C.1.* The accuracy of GPR-GAE attached to Vanilla (a) GCN, (B) GAT, (c) GPRGNN, (d) APPNP in Cora dataset against adaptive PRBCD attack for budgets $\epsilon = 0$ (clean), $\epsilon = 0.1$, $\epsilon = 0.25$, $\epsilon = 0.5$ using various attack loss types.

Given the differing mechanisms of adversarial purification and adversarial training methods in defending against attacks, the optimal hyperparameters for attacks may vary between these approaches. To determine effective attack hyperparameters for each defense strategy, we conduct a grid search across eight loss types and block sizes of [10K, 50K, 250K], under an attack budget of $\epsilon = 0.5$. For adversarial purification methods, we select vanilla GPRGNN combined with our GPR-GAE as the representative model. For non-GPR-GAE methods, we select the adversarially trained (PRBCD) GPRGNN as the representative model.

**Attack Loss Type.** Figure C.1 presents the accuracy of four GNN models combined with GPR-GAE's adversarial purification method across eight loss types with a block size of 10K on the Cora dataset. In contrast to Gosch et al. (2023), where the tanhMargin loss type is used for attacks, the Margin loss type consistently delivers the most effective attack performance across varying attack budgets on GPR-GAE. Consequently, we use the Margin loss for adaptive attacks on GPR-GAE and the tanhMargin loss for attacks on other methods.

**Block Size.** For the PRBCD attack, we use a block size of 50K for PubMed and 10K for the other datasets across all defense methods. For the LRBCD attack on Cora, we use a block size of 10K on GPR-GAE while using a block size of 250K on the rest of the methods. Additionally, for a large-scale dataset, OGB-arXiv, we exceptionally use a block

size of 3 million.

## D. Visualization of the Learned GPR Coefficients

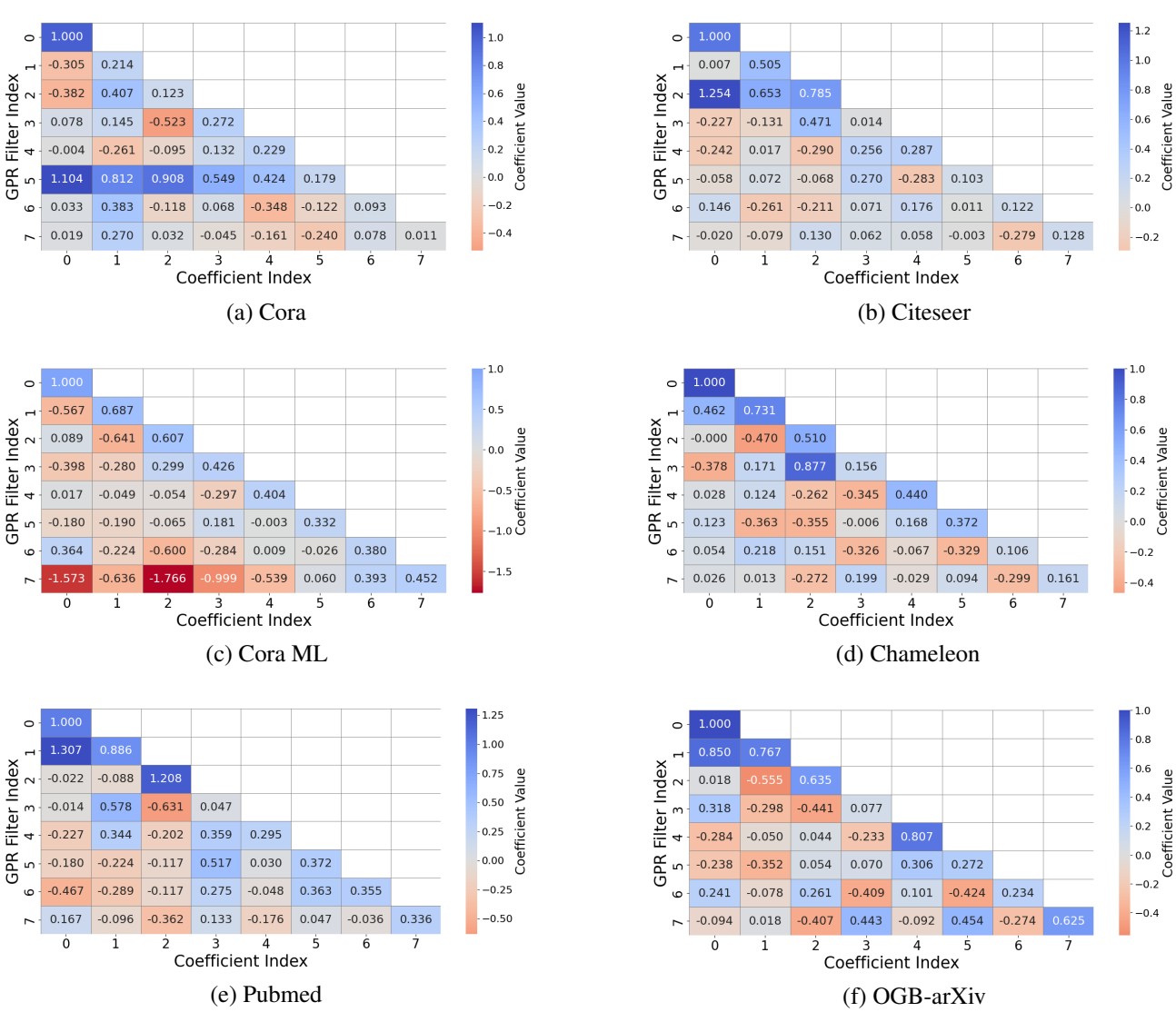

(a) Cora

(b) Citeseer

(c) Cora ML

(d) Chameleon

(e) Pubmed

(f) OGB-arXiv

*Figure D.2.* Visualization of the learned coefficients for each GPR filter in GPR-GAE when trained for adversarial purification. For the coefficient value $\gamma_{i,j}$, $i$ indicates the GPR Filter Index ($i$-th GPR Filter) and $j$ indicates the Coefficient Index (for $j$-th hop). We adjust the sign of the values so that the last coefficient values of each GPR filter are positive.

In Figure D.2, we present the learned GPR coefficients for each of the GPR filters in GPR-GAE throughout the experiments when $K = 7$. The coefficient $\gamma_{i,j}$ controls the propagation of neighborhood information $j$-hops away for the $i$-th GPR filter, allowing for fine-grained adjustment of neighborhood contributions. In Chien et al. (2021), the GPR coefficients in GPRGNN exhibit consistent signs in homophilic graphs, while fluctuating in heterophilic graphs like Chameleon, which have more complex graph structures. In contrast, the learned GPR coefficients in GPR-GAE generally display greater diversity regardless of their homophilic or heterophilic nature, efficiently regulating the inclusion or exclusion of neighborhood information. This enables the model to form distinct and unique neighborhood representations for each GPR filter with different neighborhood sizes. Furthermore, the coefficients demonstrate clear distinctions across various datasets, highlighting GPR-GAE's ability to adaptively learn appropriate neighborhood representations for each dataset, making it a

powerful data-driven approach.

# E. Algorithms

---

**Algorithm 1** Training of GPR-GAE

---

1: **Input:** Graph $G = (\mathbf{A}, \mathbf{X})$, perturbation budget $(p, q, \eta)$, model parameters $\theta$, maximum epochs, validation data $\{G_{\text{val}}\}$
2: **Output:** Trained GPR-GAE purifier $f_\theta$
3: Initialize parameters $\theta$
4: Set best_val_metric $= -\infty$
5: **for** epoch = 1 to maximum epochs **do**
6:      Sample perturbed graph $G' = (\mathbf{A}', \mathbf{X})$ from $\mathcal{B}(G, (p, q, \eta))$
7:      Predict adjacency matrix $\hat{\mathbf{A}} = f_\theta(G')$
8:      Compute total loss $\mathcal{L}$
9:      Update parameters $\theta$ using gradient descent:
$$\theta \leftarrow \theta - \nabla_\theta \mathcal{L}$$
10:      Evaluate validation metric val_metric on $\{G_{\text{val}}\}$
11:      **if** val_metric > best_val_metric **then**
12:          Update best_val_metric $\leftarrow$ val_metric
13:          Save model parameters $\theta^* \leftarrow \theta$
14:      **end if**
15: **end for**
16: **Return:** Best trained purifier $f_\theta$ with parameters $\theta^*$

---

**Algorithm 2** Multi-Step Purification with GPR-GAE

---

1: **Input:** Perturbed graph $G' = (\mathbf{A}', \mathbf{X})$, trained purifier $f_\theta$, step size $\alpha$, terminal threshold $\tau$, GNN classifier $f_\psi$
2: **Output:** Node predictions from classifier $f_\psi$
3: Initialize $\mathbf{A}^{(0)} = \mathbf{A}'$
4: **for** $t = 0$ **to** max_steps$-1$ **do**
5:      Predict purified adjacency matrix $\hat{\mathbf{A}}^{(t)} = f_\theta(\mathbf{A}^{(t)}, \mathbf{X})$
6:      Compute purification direction:
$$\Delta\mathbf{A}^{(t)} = \hat{\mathbf{A}}^{(t)} - \mathbf{A}^{(t)}$$
7:      Update adjacency matrix:
$$\mathbf{A}^{(t+1)} = \mathbf{A}^{(t)} + \alpha \cdot \Delta\mathbf{A}^{(t)}$$
8:      **if** $\frac{\|\Delta\mathbf{A}^{(t)}\|}{\|\mathbf{A}^{(t)}\|} \leq \tau$ **then**
9:          Break
10:      **end if**
11: **end for**
12: Set final purified adjacency matrix $\mathbf{A}^*$ as the last updated adjacency matrix
13: Pass $\mathbf{A}^*$ to the GNN classifier $f_\psi$
14: **Return:** Node predictions $\hat{Y} = f_\psi(\mathbf{A}^*, \mathbf{X})$

---

# F. Ablation Studies

*Table F.2.* Variations of GPR-GAE and their explanations

| Variation | Explanation |
|---|---|
| GPR-GAE | The original version of the model we use throughout the experiments. |
| w/o multi-step purification | Single-step adjacency matrix prediction, with discretization of edges using threshold 0.1. |
| w/o GPR filters | We use a 2-layered GCN for the node encoder. |
| with self-loops | Instead of $\tilde{\mathbf{A}}_{ns}$, we use $\tilde{\mathbf{A}}_s$, the normalized adjacency matrix with self-loops. |
| w/o edge reweight | Among the three perturbation methods in training, we remove the edge reweight process. |
| smaller training perturbation budget | reduced edge injection ratio $p = 1.5$ to 0.5, smaller training perturbation sample space. |
| w/o higher order dependencies | Reduced $K = 7$ to $K = 2$, with smaller neighborhood boundary coverage. |

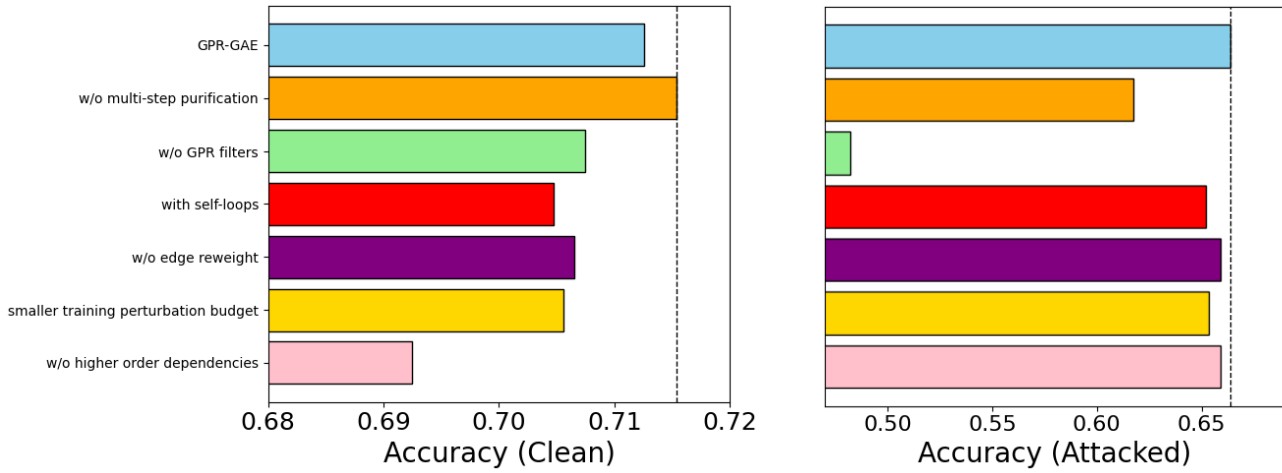

*Figure F.3.* Comparison of test accuracy for variations of GPR-GAE on Citeseer. The left figure shows accuracy under clean conditions, while the right figure illustrates accuracy under the non-adaptive PRBCD attack on a vanilla GCN.

In this section, we conduct ablation studies regarding GPR-GAE. We use vanilla GCN as the classifier, attached with the variations of GPR-GAE in Table F.2, and compare accuracy under clean and transfer attacks (PRBCD, $\epsilon = 0.5$) from vanilla GCN. Figure F.3 shows that, in general, the original variation of GPR-GAE that we use throughout the experiments achieves the best overall results. Exceptionally for the variation with no multi-step purification, it achieves a better clean accuracy compared to the original. However, under adversarial scenarios, it experiences a relatively large drop of accuracy compared to the original variation with multi-step purification, showcasing the effectiveness of the continuous and gradual graph refinements against severe perturbations.

In the case of the variation without GPR filter, which indicates replacing the node encoder with a two-layer GCN, it performs poorly, especially under adversarial attacks. This shows that the inevitable smoothing effect coming from the static neighborhood propagation scheme makes it highly ineffective under attacks, falling short in distinguishing between the clean and the adversaries. The original variation shows better performance compared to the rest of the other variations, benefiting from the non-self looped normalized adjacency matrix with more precise and distinctive control over each neighborhood propagation, learning more generalized purification directions with the edge reweight method, learning purification directions over a broader sample space using a relatively large perturbation budget, and the higher order dependencies from the larger neighborhood boundary coverage.

## G. Empirical Validation of Structural Encoding

To further extend the motivation presented in Section 3.3 and validate GPR-GAE's structural encoding capabilities, we conducted experiments to identify existent **1-hop** and **2-hop connections** in inductive settings. This evaluation compares the

GPR-GAE node encoder against three baseline node encoders: GCN, GraphSAGE, and GIN. For each node encoder, two separate edge encoders and decoders are assigned, focusing respectively on either 1-hop connections or 2-hop connections.

In 1-hop prediction, positive samples are direct connections, while negatives are random non-connections. This task primarily evaluates the model's ability to capture proximity between node pairs. In 2-hop prediction, positives are two-hop paths, and negatives are direct connections without two-hop paths. This task goes beyond simple proximity, requiring the model to distinguish between the two types of relationships that are both proximal.

**Task Definitions**

- **1-Hop Prediction:** Distinguish between $C_{1+}$ and $C_{1-}$

    - *Positive Samples* ($C_{1+}$): Direct connections in the graph.
    - *Negative Samples* ($C_{1-}$): Randomly selected non-connections of the same size as the positives.

- **2-Hop Prediction:** Distinguish between $C_{2+}$ and $C_{2-,\text{dir}}$

    - *Positive Samples* ($C_{2+}$): Connections formed by two-hop paths in the graph.
    - *Negative Samples* ($C_{2-,\text{dir}}$): Direct connections that are not two-hop.
    - *Negative Samples 2* ($C_{2-}$): Randomly selected non-two-hop connections of the same size as the positives.

**Setup**

- **Inductive Split:** Nodes were split into 80% for training, 10% for validation, and 10% for testing. Connections in validation ($C_{\text{val}}$) and testing ($C_{\text{test}}$) included at least one validation or test node, ensuring these connections were not exposed during training.

- **Training:** The node encoder and both edge encoder-decoder pairs are trained jointly as part of a unified model. We use ADAM optimizer with a learning rate of 0.01 and a weight decay of 0.00005.

    - *First Edge Encoder/Decoder:*
        * Positive train connections ($C_{1+}$): Direct connections in the training graph.
        * Negative train connections ($C_{1-}$): Random non-connections in the training graph.
        * Loss: Binary Cross-Entropy (BCE) loss on $C_{1+}$ and $C_{1-}$.
    - *Second Edge Encoder/Decoder:*
        * Positive train connections ($C_{2+}$): Two-hop connections in the training graph.
        * Negative train connections ($C_{2-,\text{dir}}$): Direct connections that are not two-hop.
        * Negative train connections 2 ($C_{2-}$): Random non-two-hop connections.
        * Loss: Combined BCE loss on $C_{2+}$, $C_{2-,\text{dir}}$, and $C_{2-}$.

- **Validation:**

    - Validation connections ($C_{1+,\text{val}}, C_{1-,\text{val}}, C_{2+,\text{val}}, C_{2-,\text{dir, val}}$) are similarly defined but included at least one validation node in each connection.
    - Performance was evaluated by summing the AUC across both 1-hop: ($C_{1+,\text{val}}, C_{1-,\text{val}}$) and 2-hop: ($C_{2+,\text{val}}, C_{2-,\text{dir, val}}$).
    - The model with the highest combined score was selected.

- **Testing:**

    - Test connections ($C_{1+,\text{test}}, C_{1-,\text{test}}, C_{2+,\text{test}}, C_{2-,\text{dir, test}}$) followed the same definitions, with at least one test node in each connection.
    - Performance was evaluated by the AUC across both 1-hop: ($C_{1+,\text{test}}, C_{1-,\text{test}}$) and 2-hop: ($C_{2+,\text{test}}, C_{2-,\text{dir, test}}$). In particular, for the 2-hop prediction task, the objective is to evaluate the model's ability to differentiate between two types of connections, both of which are considered proximal, as they connect nodes within a relatively short range of 2-hops.

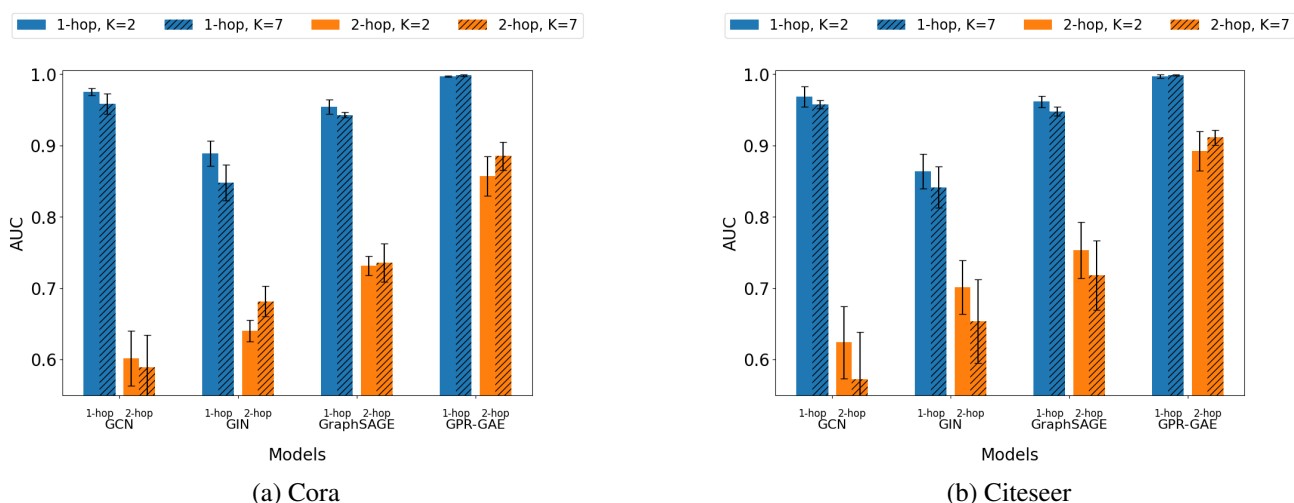

(a) Cora

(b) Citeseer

*Figure G.4.* AUC for existent 1-hop and 2-hop link identification using different GAE models

## Results

Results in Figure G.4 demonstrate that GPR-GAE achieves the highest AUC scores in both 1-hop and 2-hop prediction tasks for $K = 2$ and $K = 7$. Unlike traditional GAE architectures, GPR-GAE exhibits a consistent performance improvement as $K$ increases, showcasing its ability to fully leverage higher-order dependencies while mitigating oversmoothing. Moreover, GPR-GAE significantly outperforms traditional GAEs in the 2-hop prediction task. This highlights the advantage of its unique multiscale neighborhood representations, which are derived from distinctively stored outputs of GPR filters. These representations enable GPR-GAE to better model complex relationships, such as $C_{2+,\text{test}}$ and $C_{2-,\text{dir, test}}$, moving beyond simply encoding proximal connections. Consequently, the demonstrated results form a robust foundation for tasks such as adversarial purification that require advanced structural capabilities.

