# OpenReview forum: "Self-supervised Adversarial Purification for Graph Neural Networks"
_ICML.cc/2025/Conference — ICML 2025 poster_

### Official Review · Reviewer_BjpG · 2025-03-04

**Overall Recommendation:** 3

**Summary:**

The paper proposes a method to defend GNNs that is based on a separate GNN classifier and a purifier. The main idea is to decouple the classifier and purifier and learn a multi-step purifier using generalized pagerank. Extensive experiments are provided showing that this approach outperforms state-of-the-art defenses or purification methods.

**Claims And Evidence:**

Most of the major claims are well supported. However, I feel some claims are rather vaguely supported / evidenced.

**(I)** They claim to contribute a decoupling into two components: a classifier for accuracy and a purifier to restore graph structure. But aren't the static approaches such as Jaccard-GCN or SVD-GCN also already "decoupled" in the sense of first preprocessing the graph and then applying a GNN on top?
**(II)** In the abstract, the method is framed as a solution to overcome the robustness-accuracy tradeoff. While empirically the trade-off seems slightly better than in the compared to methods, claiming a perfect balance of both would require more extensive also theoretic study.

**Essential References Not Discussed:**

I do think that all essential references are discussed.

**Experimental Designs Or Analyses:**

Yes, all experiments/analyses in the main draft.

## Strengths

* Follow state-of-the-art evaluation protocols for evasion attacks (e.g., inductive setting)
* I really like Table 1. The number of experiments and comparisons per dataset are impressive. They clearly show the effectiveness of GPR-GAE compared to adversarial training and other robust GNNs.
* Ablation study on the many design choices provided.
* Code provided.

## Weaknesses

**(I)** As all homophilic datasets are citation datasets, I would be curious about the performance on a homophilic non-citation datasets e.g. WikiCS.
**(II)** Given Cora is the main dataset for a detailed comparison between methods in the main draft, I would have expected Table 2 to include the Cora dataset (or that other purification methods have also been employed on Cora).
**(III)** All results are presented with standard deviations but I can't find a description about how many random seeds have been used.

**Methods And Evaluation Criteria:**

Yes.

**Other Comments Or Suggestions:**

See questions.

**Other Strengths And Weaknesses:**

## Weaknesses

* The method seems to have many hyperparameters, e.g. $\tau, p, q, \eta, \delta$, #purification steps and other design decisions. However, the authors provide ablations.

**Questions For Authors:**

* Is the edge injection step a uniform sampling from the complementary edge set? Furthermore, please make explicit how you sample from $\mathcal{E}^C$.
* Make explicit if the $L_\text{restore}$ loss is maximized or minimized. Furthermore, as far as I understood how $L_\text{restore}$ is currently defined in the text it should be maximized so as to make sense (minimization would be trivial bei outputting an $A_{ij}$-prediction around $0$ for an $\{i,j\} \in \mathcal{E}$ and that is exactly not what we want). However, $L_\text{sym}$ should be minimized. Thus, $L_\text{restore}+\delta L_\text{sym}$ doesn't make sense with the current definitions. Is there a sign-error in the current presentation?

**Relation To Broader Scientific Literature:**

The work contributes a superior purification method compared to previous static approaches. Furthermore, they show that their approach outperforms state-of-the-art defenses.

**Theoretical Claims:**

I checked the correctness of all theoretical claims and went through the proofs in the Appendix.

**Issues**

**(I)** Sec 2.2 one could be explicit about what the expectation over is meant. E.g. the general objective is to minimize the expected loss over the distribution of graphs that could be sampled conditioned on the already given graph. However, for Proposition 3.1 to hold e.g., the empirical risk is used, as this is the one that can actually be used for training a model on a concrete dataset. This is no criticism that the empirical risk is used, just that one could be explicit in the main draft.
**(II)** Given a fixed perturbation Proposition 3.1 holds true. However, during the attack generation for adversarial training, each iteration will have different $\mathcal{V}_\text{unaffected}$ and $\mathcal{V}_\text{affected}$. Thus, while I do understand the motivation for the approach, I'm not 100% convinced of the argumentation. But as it is "just" a motivation, this may be enough.
**(III)** Could you clarify the Lipschitzness of your proposed updating scheme? Crucially, Thm. 3.2 requires $L < 1$.

---

> ### Author Rebuttal · Authors · 2025-04-01
>
> Thank you for your thoughtful and constructive comments. We would like to address your concerns (C) with the following responses:
>
> ---
>
> > **C1.** Aren't the static approaches such as Jaccard-GCN or SVD-GCN also already "decoupled"?
>
> Unlike Jaccard-GCN and SVD-GCN, which apply fixed heuristics for preprocessing, our contribution explicitly motivates **decoupled learning**: the purifier (GPR-GAE) independently learns graph reconstruction via a self-supervised objective, while the classifier focuses solely on accuracy. Thus, we advocate decoupling not merely in preprocessing steps, but crucially within the learning process itself.
>
>
> ---
> > **C2.** Claiming a perfect balance of both trade-off terms would require more extensive also theoretic study.
>
> We thank the reviewer for highlighting this point. To clarify, when we say "overcome" in abstract, we refer to the constraints of traditional methods intertwining accuracy and robustness within one classifier. Proposition 3.1 theoretically decomposes these conflicting objectives, motivating our design of specialized modules that independently handle each—especially robustness. While we fully agree that claiming a perfect balance requires further theoretical study, our empirical results show a substantial improvement in **mitigating** the trade-off. We will ensure this nuance is clearly articulated in future revisions.
>
> ---
> > **C3.** Sec 2.2 one could be explicit about what the expectation over is meant.
>
> We thank the reviewer for the advice and will explicitly state that the expectation in Sec 2.2 refers to the empirical risk over nodes in the given graph.
>
> ---
> > **C4.** Given a fixed perturbation Proposition 3.1 holds true. However, during the attack generation for adversarial training, each iteration will have different $V_{\text{unaffected}}$ and $V_{\text{affected}}$.
>
> We thank the reviewer for this insightful remark. To clarify the notation, we will explicitly denote the node sets as $V_{\text{unaffected}}^{G'}$ and $V_{\text{affected}}^{G'}$ to reflect their dependence on each perturbation $G'$ generated during adversarial training. Importantly, this notation refinement does not affect our core claim: each training iteration still highlights the inherent tension between accuracy and robustness, reinforcing our rationale for decoupling these conflicting objectives into separate specialized modules.
>
> ---
> > **C5.** Could you clarify the Lipschitzness of your proposed updating scheme? Crucially, Thm. 3.2 requires $L < 1$.
>
> This is a valuable question that we are happy to clarify. The Lipschitz condition $L < 1$, required by Theorem 3.2 for convergence of the purification update scheme, is encouraged by our training objective: minimizing $\mathcal{L}(\theta) = \ell(f_{\theta}(G'), G)$, where $G'$ is a perturbed version of the clean graph $G$. Since $G$ is fixed during training and the model learns to satisfy $f_{\theta}(G) \approx G$, the loss encourages the condition $\||f_{\theta}(G') - f_{\theta}(G)\|| < \||G' - G\||$, i.e., $f_{\theta}$ behaves as a contraction mapping around $G$, approximating a Lipschitz constant $L < 1$ under a sufficient perturbation budget. This ensures stable and convergent updates in the multi-step purification process, as required by Theorem 3.2.
>
> ---
> > **C6.** Additional experiment on WikiCS and Cora (Table 2)
>
> | Model | WikiCS (clean/0.25/0.5) | Cora (clean/0.25/0.5) |
> |--|-|-|
> | GCN (Vanilla) |75.0 / 29.0 / 24.5| 79.4 / 46.9 / 29.3|
> | EvenNet  | **76.7** / 35.8 / 31.9| **81.4** / 65.9 / 56.4 |
> | SoftMedianGDC| 73.5 / 33.5 / 30.7 | 77.4 / 62.3 / 53.6|
> | Jaccard-GCN | 73.4 / 58.8 / 52.5 | 78.5 / 64.1 / 50.9 |
> | SVD-GCN | 73.0 / 42.3 / 36.9 | 77.2 / 64.8 / 50.2 |
> | GOOD-AT | 74.1 / 58.7 / 51.2 | 77.7 / 65.4 / 61.0 |
> | **$GPR\text{-}GAE_{GCN\text{-}Vanilla}$** |75.1 / **66.8 / 63.0** | 79.4 / **75.1 / 72.6** |
>
> ---
> > **C7.** How many random seeds?
>
> We used 5 random seeds across all experiments.
>
> ---
> > **C8.** Seems to have many hyperparameters.
>
> Although our method involves several hyperparameters, we deliberately keep most of them fixed across all datasets, demonstrating that our approach generalizes well without extensive tuning.
>
> ---
> > **C9.** Is the edge injection step a uniform sampling from the complementary edge set?
>
> Yes, the edge injection step involves uniform sampling from the set of non-existent edges $\mathcal{E}^C$. Specifically, we first calculate the number of edges to inject as $|\mathcal{E}_{\text{inject}}| = p \cdot |\mathcal{E}|$, and then uniformly sample this number of edges from $\mathcal{E}^C$ for injection.
>
>
> ---
> > **C10.** Make explicit if the $L_{restore}$ is maximized or minimized.
>
> Thank you for the correction — you are absolutely right. The current loss formulation is misleading as it implies maximization, which contradicts our intended objective of minimizing the restoration loss. We appreciate your correction, and we will adjust the sign and clarify the explanation in the revision.

---

> > ### Comment · Reviewer_BjpG · 2025-04-04
> >
> > I thank the authors for the rebuttal. It addressed most of my concerns and thus, I'm increasing my score. I think it would be particularly valuable to include a discussion on why the assumptions on Thm 3.2 should be satisfied by the method into the camera-ready version (next to the other points in the rebuttal).

---

> > > ### Author Response · Authors · 2025-04-05
> > >
> > > We sincerely thank the reviewer for their thoughtful reconsideration and constructive feedback. We appreciate the suggestion regarding Theorem 3.2 and agree that a discussion of its assumptions would help improve the clarity of the work. We will make sure to incorporate this point alongside the other clarifications raised during the review process.
> > >
> > > If there’s anything else we can clarify or expand on, we’d be more than happy to help.

---

### Official Review · Reviewer_ShLW · 2025-03-10

**Overall Recommendation:** 2

**Summary:**

This paper studies the robustness of GNNs against adversarial attacks from the perspective of adversarial purification. The authors introduce a self-supervised adversarial purification framework that preprocesses input data to remove adversarial perturbations before classification. Experimental results on a wide range of graph datasets demonstrate that the proposed method achieves state-of-the-art robustness while maintaining high accuracy. Overall, the proposed approach offers a promising direction for improving adversarial robustness in GNNs without compromising classification accuracy.

**Claims And Evidence:**

The claims made in the submission are generally well-supported by convincing evidence.

**Essential References Not Discussed:**

N/A

**Experimental Designs Or Analyses:**

See Weaknesses and Questions below.

**Methods And Evaluation Criteria:**

There are missing comparison with adversarial training based graph defense methods

**Other Comments Or Suggestions:**

N/A.

**Other Strengths And Weaknesses:**

Pros:

- The motivation of defending GNNs against adversarial attacks using a purification module is reasonable.
- The idea of decoupling the learning objective of adversarial training into accuracy and robustness terms is interesting and technically sound.
- Experimental results demonstrate the effectiveness of the proposed method compared to baselines.



Cons:

- Many adversarial training methods are cited but not compared in the experiments.
- Larger datasets, such as OGBN-MAG or OGBN-Products, are needed to demonstrate the scalability of the proposed defense.
- Although the proposed method can serve as a plug-and-play purification module for various GNN architectures, it is expected to introduce additional computational overhead. The authors should discuss the complexity of the proposed method to evaluate whether it is justified as a purification module.

**Questions For Authors:**

- The proposed method is effective against evasion attacks, but how does it perform against poisoning attacks?

**Relation To Broader Scientific Literature:**

N/A

**Theoretical Claims:**

The claim in Proposition 3.1 is clear. However, according to Proposition 3.1, with a deep GNN (e.g., when the number of layers ≥ 3), most nodes in the graph will be affected. My question is, in such a case, does the robustness term become the dominant term in the loss function? If so, will this affect the model’s accuracy?

---

> ### Author Rebuttal · Authors · 2025-04-01
>
> Thank you for your thoughtful and constructive comments. We would like to address your concerns (C) with the following responses:
>
> ---
>
> > **C1.** The claim in Proposition 3.1 is clear. However, according to Proposition 3.1, with a deep GNN (e.g., when the number of layers ≥ 3), most nodes in the graph will be affected. My question is, in such a case, does the robustness term become the dominant term in the loss function? If so, will this affect the model’s accuracy?
>
> As you noted, with increased depth, more nodes are affected by perturbations, which could overemphasize the robustness term in conventional adversarial training. This overemphasis can indeed lead to a drop in accuracy due to the model learning mostly from fundamentally corrupted data. This consideration is precisely why we advocate for decoupling accuracy and robustness in our self-supervised adversarial purification framework. The decoupling ensures neither objective overshadows the other, preventing performance degradation. We appreciate you highlighting this important architectural consideration!
>
> ---
>
> > **C2.** Many adversarial training methods are cited but not compared in the experiments.
>
> To the best of our knowledge, [1] is the most recent published work on adversarial training in Graph Neural Networks (GNNs). Notably, it is the first to explicitly address the validity-related limitations of previous adversarial training in the transductive setting, as mentioned in Section 2.1. We therefore focus our comparison on this state-of-the-art approach.
>
> | Model | Cora | Citeseer |
> |---|-|---|
> | $S^2GC$ [2] |52.3±3.4|50.2±3.6|
> | $GPRGNN$ [1] |71.4±3.0|65.7±4.4|
> | $GPR\text{-}GAE_{GCN\_{Vanilla}}$ | 75.1±2.7|68.7±3.1|
>
> The table further supports our choice of baseline by reporting the performance of $GPR\text{-}GAE_{GCN\_Vanilla}$ and [1] with GPRGNN as backbone under a non-adaptive PRBCD attack with $\epsilon = 0.25$ in an inductive setting. We also include a previous adversarial training method [2], evaluated with its best-performing backbone, $S^2GC$. Although [2] demonstrates strong results against other adversarial training methods under the transductive setting in its original paper, it significantly underperforms compared to the adversarial training baseline we use [1] under our inductive setup.
>
> ---
>
> > **C3.** Larger datasets, such as OGBN-MAG or OGBN-Products, are needed to demonstrate the scalability of the proposed defense.
>
> We follow prior works on adversarial robustness, which commonly use OGBN-Arxiv as the largest-scale benchmark. For further demonstration, we evaluate the scalability of GPR-GAE on OGBN-Products, which has over 10× more nodes and 50× more edges than OGBN-Arxiv. Due to PRBCD and LRBCD attacks running out of memory on OGBN-Products, we use DICE [3], a non-gradient based attack, to demonstrate the scalable robustness of GPRGAE in the following table.
>
> |Model|Clean| $ \epsilon = 0.1 $ | $ \epsilon = 0.25 $ | $ \epsilon = 0.5 $ |
> |-|-|-|-|-|
> | $ GCN\_{Vanilla} $ |73.7±1.8|68.3±2.4|62.7±2.0|55.9±2.6|
> | $GPR\text{-}GAE_{GCN\_{Vanilla}}$ |73.4±2.3|71.5±3.1|69.7±2.5|68.4±2.4|
>
> The results indicate that GPR-GAE not only scales efficiently to significantly larger datasets like OGBN-Products but also maintains robust performance under adversarial conditions.
>
> ---
>
> > **C4.** Although the proposed method can serve as a plug-and-play purification module for various GNN architectures, it is expected to introduce additional computational overhead. The authors should discuss the complexity of the proposed method to evaluate whether it is justified as a purification module.
>
> Our multi-step purification process introduces at most a linear overhead, regardless of the GNN classifier architecture used. As detailed in Section 5.4 and Section B, while a K-layer GCN has a node encoding complexity of $O(K · |E| · Z)$, our purification module incurs an extra $O(K · |E| · Z^2)$ cost over a small fixed number of purification steps. This additional cost scales linearly with the number of edges and,in our view, is justifiable by the substantial improvements in adversarial robustness achieved by our method across diverse classifers. For empirical evidence, please refer to our response to Reviewer "kPxh" (C4).
>
>
> ---
> > **C5.** The proposed method is effective against evasion attacks, but how does it perform against poisoning attacks?
>
> Thank you for raising this important point. Our methodology focuses on evasion attacks, as our aim is to address theoretical limitations of adversarial training, which is primarily explored in the evasion setting. While GPR-GAE is not explicitly designed for poisoning, its self-supervised, model-agnostic framework offers potential for extension—an exciting direction for future work.
>
> ---
>
> [1] Adversarial Training for Graph Neural Networks: Pitfalls, Solutions, and New Directions
>
> [2] Spectral Adversarial Training for Robust Graph Neural Network
>
> [3] Adversarial Attacks on Graph Neural Networks via Meta Learning

---

### Official Review · Reviewer_BFFi · 2025-03-13

**Overall Recommendation:** 2

**Summary:**

Traditional defense strategies for Graph Neural Networks (GNNs), such as adversarial training, often struggle to balance accuracy and robustness, as they entangle these competing objectives within a single classifier. This paper challenges that approach and introduces a novel self-supervised adversarial purification framework designed to decouple robustness from classification. By incorporating a dedicated purifier that preprocesses input data before classification, the proposed method ensures enhanced resilience against adversarial attacks. Experimental results across multiple datasets and attack scenarios highlight its state-of-the-art robustness, positioning it as an adaptable, plug-and-play solution for fortifying GNN classifiers.

## update after rebuttal

The primary concern with this paper is its novelty. Prior to the rebuttal, my stance leaned toward a Weak Reject. During the rebuttal phase, the authors provided some reasonable evidence to support their contributions. However, I still have remaining concerns regarding the novelty of the work. Taking these points into account, I would like to maintain a **Neutral position**. I do **not lean strongly toward either acceptance or rejection during the AC and reviewer discussion phase.**

**Claims And Evidence:**

Yes.

**Essential References Not Discussed:**

See Other Strengths And Weaknesses.

**Experimental Designs Or Analyses:**

See Other Strengths And Weaknesses.

**Methods And Evaluation Criteria:**

Yes.

**Other Comments Or Suggestions:**

See Other Strengths And Weaknesses.

**Other Strengths And Weaknesses:**

### Pros

- The paper addresses an important problem with a clear motivation and concrete analysis.
- The writing is well-structured and clear.
- Extensive experiments support the effectiveness of the proposed approach.

---

### Cons

- Concerns regarding novelty
   - The core idea of separating the graph purifier and classifier under an evasive setting closely resembles test-time graph transformation [1].
   - The proposed GPR-GAE lacks technical novelty. Using GAE as a noisy edge predictor is a well-established approach [2,3], and simply replacing the GCN layer with GPR-GNN does not introduce significant innovation.
   - I strongly encourage the authors to discuss these concerns in detail.
- Lack of baseline comparisons
   - The paper should include comparisons with [1,2,3]. Additionally, several adversarial training-based robust GNN methods, such as [4], should be considered as baselines.

- Lack of qualitative analysis
   - Providing qualitative results demonstrating how the proposed method enhances performance would be beneficial.

---

### References
[1] Empowering Graph Representation Learning with Test-Time Graph Transformation

[2] How to Find Your Friendly Neighborhood: Graph Attention Design with Self-Supervision

[3] Self-Guided Robust Graph Structure Refinement

[4] Adversarial Graph Contrastive Learning with Information Regularization

**Questions For Authors:**

See Other Strengths And Weaknesses.

**Relation To Broader Scientific Literature:**

Contributing to various scientific GNN applications.

**Theoretical Claims:**

No discussions.

---

> ### Author Rebuttal · Authors · 2025-04-01
>
> Thank you for your thoughtful and constructive comments. We address your concerns (C) as follows:
>
> ---
>
> > **C1.** The core idea closely resembles test-time graph transformation [1].
>
> We understand the concerns regarding resemblence with TTGT[1] and wish to highlight four key differences:
>
> 1. Dedicated Parameter Training
> TTGT leverages embeddings from a pretrained classifier and **avoids additional training**. It performs test-time transformation by optimizing a surrogate loss tied to the classifier via Projected Gradient Descent-an approach that closely resembles PGD-style attacks. In contrast, **our method introduces a dedicated purification module trained on the training graph**, which **performs purification via learned inference, independently of the classifier**.
>
> 2. Purification Steps
> Our framework performs **multi-step, continuous purification**, enabling gradual refinement and better resilience to severe perturbations. TTGT, by comparison, uses a **single-step discrete projection**. Our iterative purification process differs not only from TTGT but also from other purification approaches.
>
> 3. Theoretical Focus
> TTGT minimizes a surrogate loss approximating class-conditional entropy H(Z|Y). Its guarantee (Theorem 2 in [1]) assumes that class-conditional means ($c_k$) approximate the true distribution—**an assumption that breaks under adversarial attacks**, which distort feature distributions. Hence, **TTGT is more suitable for mild domain shifts, as reflected in the nature of follow-up works that cite it**.
>
> 4. Purification Budget
> **TTGT uses a fixed edge-flip budget** like attacks, limiting its adaptability. In contrast, **our method learns adaptive, continuous multi-step updates without a predefined budget**, improving robustness across varying attack strengths.
>
> ---
>
> > **C2.** GPR‑GAE lacks technical novelty, Using GAE as a noisy edge predictor is well‑established [2,3].
>
> GPR-GAE introduces both **architectural and conceptual novelties** over GPRGNN. While GPRGNN uses a single set of learned propagation coefficients for supervised classification, GPR-GAE employs multiple distinct GPR filters, each with its own propagation weights, concatenated to capture **diverse multi-scale neighborhood patterns**. We also exclude self-loops to better highlight meaningful structural signals. These design choices explicitly aim to enhance the model’s structural capability to distinguish clean from adversarial structures, as empirically validated in Figure G.4.
>
> Unlike prior GAE-based defenses such as [3, 5], which rely on additional heuristic pre-processing (e.g., feature similarity) before applying GAE based denoising like [2], GPR-GAE is **fully data-driven** with its enhance structural capabilities, learning to purify adaptively without relying on such assumptions.
>
> Moreover, while most existing self-supervised defense methods like [3, 5] focus on the poisoning setting, our work addresses the underexplored area of self-supervised defense in the evasion setting with different challenges—applying a targeted self-supervised framework to this context and introducing an additional point of novelty.
>
> Finally, although GPR-GAE utilizes similar loss as the conventional GAEs, its core focus lies in **learning purification directions in a continuous space** via edge re-weighting, effectively leveraging the train graph to enable more precise and generalizable purification on the test stage attacks.
>
> ---
>
> > **C3.** Comparison with [1,2,3] and other adversarial training.
>
> We acknowledge the relevance of [1,2,3] and include them in our comparison below. However, some's core assumptions differ from ours, which limits direct comparability.
>
> [3] builds on SuperGAT [2] with the rest of the methodology tailored for the poisoning problem setting. Since our work focuses on the evasion setting, we represent both [2,3] as $SuperGAT_{MX}$ [2] for comparison. Also, we evaluate $TTGT_{GCN}$ [1] using a 25% purification budget, matching the perturbation level of the non-adaptive PRBCD attack used in the following table, to reflect its best-case performance in our setup.
>
> |Model|Cora|Citeseer|
> |-|-|-|
> |$GCN$ (For comparison)|46.9±1.5|40.0±2.3|
> |$TTGT_{GCN}$ [1]|49.6±2.0|44.2±2.1|
> |$SuperGAT_{MX}$ [2,3]|54.6±4.6|46.4±1.7|
> |$GPR\text{-}GAE_{GCN}$|75.1±2.7|68.7±3.1|
>
> While [1,2,3] offer valuable contributions, their focus on different problem settings limits their robustness under our evaluation scenario.
>
> Please see our response to Reviewer "ShLW" (C2) for concerns on the adversarial training baselines.
>
> ---
>
> > **C4.** Qualitative results would be beneficial.
>
> Figure D.2 provides qualitative insights by visualizing the learned GPR coefficients, which exhibit diverse, dataset-specific propagation patterns. This highlights GPR-GAE's capabilities of adaptive purification, leading to improved performance across various datasets.
>
> ---
>
> [5] Reliable Representations Make A Stronger Defender: Unsupervised Structure Refinement for Robust GNN

---

> > ### Comment · Reviewer_BFFi · 2025-04-04
> >
> > I appreciate the authors’ efforts in addressing the concerns. While Comments C3 and C4 have been resolved, I believe the overall novelty of this work remains limited, even after considering the authors’ rebuttal. The core idea of the purify-then-train approach appears to stem from the concept of test-time transformation. Although the authors emphasize certain differences between their method and test-time transformation, these distinctions seem incremental, primarily involving the addition of technical details to an existing idea. Similarly, the novelty of GPR-GAE also appears limited. While the authors note some distinctions from prior work, these seem to amount to minor modifications built upon established approaches. Given these concerns, I will maintain my initial score.

---

> > > ### Author Response · Authors · 2025-04-04
> > >
> > > We appreciate your thoughtful response, and would like to further clarify on your still remaining concerns regarding the novelty of our work.
> > >
> > > > The core idea of the purify-then-train approach appears to stem from the concept of test-time transformation.
> > >
> > > We believe you may have meant **"purify-then-infer"** rather than **"purify-then-train"**, as the latter typically refers to purification defenses targeting poisoning attacks, which is not the focus of our work.
> > >
> > > We would also like to stress that our approach **does not stem from the concept of test-time transformation**. In fact, the notion of **"purify-then-infer"** defines the adversarial purification setting in the context of evasion attacks—that is, purifying a perturbed graph before applying a trained classifier at inference time. All adversarial purification baselines we compare against follow this same setting. As such, our work is grounded directly in this established purification paradigm, as reflected in our title: *"Self-supervised Adversarial Purification for Graph Neural Networks."*
> > >
> > > What distinguishes our approach is not the use of a purification step, but the **self-supervised learning framework** we introduce within that paradigm. Specifically, our method is designed to **decouple the learning of conflicting objectives**—classification and purification—by separating the purification model entirely from the classifier, both in terms of training and operation. This design allows the purifier to learn meaningful purification directions from the training graph itself, without relying on classifier feedback or shared representations.
> > >
> > > In contrast, test-time transformation:
> > > 1. Does **not** include a separate purification module,
> > > 2. Does **not** learn from or leverage the training graph, and
> > > 3. Directly **uses classifier embeddings** to guide transformation.
> > >
> > > Given these core differences, we respectfully but firmly disagree that our distinctions are incremental—either in comparison to test-time transformation or to existing adversarial purification methods. Our goal and core idea has been to move toward a more principled and independent framework for learning robust purification strategies and not the general "purify-then-infer" concept. We hope this clarification helps communicate that more clearly.
> > >
> > > ---
> > >
> > > >While the authors note some distinctions from prior work, these seem to amount to minor modifications built upon established approaches.
> > >
> > > We appreciate the perspective and understand that our contributions may appear as minor modifications built upon established approaches. However, we would like to clarify that while some components may seem incremental at a glance, they are in fact carefully motivated and targeted design choices aimed at addressing key limitations in existing methods. Specifically:
> > >
> > > - **The concatenated GPR filter design in GPR-GAE** enables the model to capture diverse multi-scale structural patterns, going beyond traditional autoencoders that rely on a single neighborhood aggregation for better differentiation between clean and adversarial regions.
> > > - **The multi-step purification process in continuous space** avoids abrupt, one-shot changes and allows for adaptive, iterative refinement of graph structure. This leads to significantly improved recovery with more precise purification, especially under severe perturbations.
> > > - **Our perturbed graph sampling strategy** introduces a more realistic and continuous training environment by combining edge injection, masking, and reweighting. This helps the purifier learn robust purification directions applicable across a broader spectrum of adversarial patterns.
> > >
> > > While not all components might seem individually novel, we believe the overall framework provides a meaningful step toward bridging the gap between traditional graph autoencoding approaches and the specific challenges of adversarial robustness. Our design choices are purposefully aligned to make purification more effective, adaptable, and broadly applicable in adversarial settings.
> > >
> > > ---
> > >
> > > We hope this clarification provides a clearer understanding of our motivations and contributions, and would be grateful if you could reconsider the evaluation of our work in light of these points. Thank you once again for your thoughtful and constructive feedback.

---

### Official Review · Reviewer_kPxh · 2025-03-27

**Overall Recommendation:** 3

**Summary:**

This study introduces a self-supervised adversarial purification framework to enhance the robustness of GNNs against attacks. Unlike traditional methods that merge accuracy and robustness in a single classifier, their approach (GPR-GAE) employs a dedicated purifier to cleanse input data prior to classification. Experimental results show that GPR-GAE achieves  promising results.

**Claims And Evidence:**

yes

**Essential References Not Discussed:**

[1] How does heterophily impact the robustness of graph neural networks? theoretical connections and practical implications

[2] Graph neural networks with diverse spectral filtering

[3] Node-oriented spectral filtering for graph neural networks

[4] Node-wise filtering in graph neural networks: A mixture of experts approach

[5] Node-wise localization of graph neural networks

**Experimental Designs Or Analyses:**

yes

**Methods And Evaluation Criteria:**

yes

**Other Comments Or Suggestions:**

See the weakness part in "Other Strengths And Weaknesses".

**Other Strengths And Weaknesses:**

**Strengths:**

1. The research questions addressed in the paper and the authors' designs are both well-motivated.
2. The experimental analysis, enhanced by visualization, is thorough and comprehensive.
3. The paper is well-written and easy to follow.

**Weaknesses:**

1. The authors should further elaborate on the connections to existing work on GNNs with spectral filtering (a.k.a. spectral GNNs).
2. While the authors employ a node-wise filtering approach in their method, they fail to discuss similar works extensively. It is recommended that they consider and engage with relevant literature, such as [2, 3, 4, 5], and select appropriate methods to strengthen their baseline comparisons in the experimental section.
3. This work primarily focuses on the robustness of GNNs, employing spectral filtering as a solution. However, the use of the GPR filter (adaptive filter) is closely tied to the issue of graph heterophily. The authors are encouraged to review the work cited as [1] and provide a more in-depth discussion within this context to enhance the impact of their work in the field.
4. While a complexity analysis is included, it is suggested that the authors also provide empirical results.

[1] How does heterophily impact the robustness of graph neural networks? theoretical connections and practical implications

[2] Graph neural networks with diverse spectral filtering

[3] Node-oriented spectral filtering for graph neural networks

[4] Node-wise filtering in graph neural networks: A mixture of experts approach

[5] Node-wise localization of graph neural networks

**Questions For Authors:**

See the weakness part in "Other Strengths And Weaknesses".

**Relation To Broader Scientific Literature:**

(1) graph neural networks (2) robustness (3) generalization

**Theoretical Claims:**

yes

---

> ### Author Rebuttal · Authors · 2025-04-01
>
> Thank you for your thoughtful and constructive comments regarding our work. We would like to address your concerns (C) with the following responses:
>
> ---
> > **C1.** The authors should further elaborate on the connections to existing work on GNNs with spectral filtering (a.k.a. spectral GNNs).
>
> We thank the reviewer for the suggestion. Among spectral GNNs with learnable coefficients—such as ChebNet [6], BernNet [7], and GPRGNN—we build on GPRGNN due to its expressiveness and flexibility. Unlike ChebNet and BernNet, which use fixed polynomial bases, GPRGNN leverages learnable coefficients over a monomial basis, enabling unconstrained adaptation to diverse spectral responses.
>
> We extend this by employing multiple GPR filters in parallel, each with independently learned coefficients, and concatenate their outputs. This design captures a wider range of spectral behaviors, enhancing generalization under structural perturbations.
>
> Additionally, we omit self-loops by using the normalized adjacency $\tilde{A}_{ns} = D^{-1/2} A D^{-1/2}$, which preserves spectral properties. Since self-loops suppress high-frequency components, excluding them helps retain fine-grained variations crucial for tasks sensitive to local structure. As shown in Figure F.3, this improves purification under adversarial conditions.
>
> |Filters|Cora|Citeseer|
> |-|-|-|
> |Chebnet[6]|68.3±2.8| 63.8±2.9 |
> |Bernet[7]|67.6±3.1| 62.9±3.0 |
> |GPR (Ours)|69.7±2.9|64.1±2.4|
>
> Under adaptive PRBCD attack ($\epsilon = 0.25$), our GPR-based filters show higher robustness than ChebNet and BernNet, confirming the benefit of its flexible, unconstrained design.
>
>
>
> ---
> > **C2.** The paper adopts a node-wise filtering strategy but lacks sufficient discussion of related node-wise spectral filtering methods (e.g., [2–5])
>
> We thank the reviewer for this observation. While our method shares some perspective with node-wise filtering—particularly in leveraging multiple filters to capture diverse structural signals—we emphasize that GPR-GAE takes a fundamentally different approach by using shared global filters instead of assigning unique filters per node. Recent works [2–5] propose node-wise filtering, adapting filter parameters locally based on each node’s structural context. While expressive, this approach is highly vulnerable to adversarial settings: localized attacks that perturb neighborhoods can distort the context for node-specific filter selection, leading to unreliable or misleading outcomes.
>
> In contrast, GPR-GAE employs multiple global GPR filters applied uniformly across the graph. These filters are trained to generalize over a wide range of structural patterns, offering inherent robustness to localized noise. Concatenating these globally learned filters yields a rich spectrum of responses without requiring per-node adaptation, avoiding amplification of adversarial perturbations.
>
> ---
> > **C3.** Discussion related to [1].
>
> We appreciate the reviewer’s insightful suggestion. The theoretical insights in [1] align well with our training strategy. First, [1] observes that structural attacks shift homophilic graphs toward heterophily, emphasizing the value of heterophil-aware robustness. Our self-supervised training strategy injects a large ratio of negative edges—unlike prior GAEs—exposing the model to heterophilous patterns even on homophilic graphs, and further discussed in Figure D.2. Second, [1] advocates separating ego- and neighbor-embeddings. As discussed in Section 4.1, our encoding concatenates representation $H_{\theta_k}$ from distinct GPR filters, with $H_{\theta_0}$ explicitly set to $H^{(0)}$—the initial node embedding—preserving ego-information throughout purification. This design enhances robustness by preserving self-information from potentially corrupted neighborhoods. Under adaptive PRBCD attack ($\epsilon = 0.25$), removing self-information from the concatenation results in a performance drop of $2.4$% on Cora and $1.2$% on Citeseer, validating its contribution to robustness.
>
> ---
> > **C4.** While a complexity analysis is included, it is suggested that the authors also provide empirical results.
>
> | Model | Train(per epoch) | Inference($ \epsilon=0.25 $) |
> |-|-|-|
> | GOOD-AT | 2744.9ms | 16.2ms |
> | GPR-GAE | 65.6ms | 87.2ms |
>
> The table compares GPR-GAE with the recent purification baseline, GOOD-AT, on Cora, reporting average training time per epoch and inference time under PRBCD attack ($\epsilon = 0.25$) using GCN as the classifier. GPR-GAE trains much faster (65.6 ms vs. 2744.9 ms). While its inference time (87.2 ms) is higher, it remains practical. GOOD-AT’s cost stems from adversarial sample generation. For reference, inference with standalone GCN takes 8.5 ms, showing GPR-GAE adds minimal overhead for substantial robustness gains.
>
> ---
>
> [6] Convolutional Neural Networks on Graphs with Fast Localized Spectral Filtering
>
> [7] BernNet: Learning Arbitrary Graph Spectral Filters via Bernstein Approximation

---

### Decision · Program_Chairs · 2025-05-01

**Decision:**

Accept (poster)

**Comment:**

The proposed approach is well-motivated. The main ideas (purification, decoupling) are interesting and sound. Importantly, as pointed out by Reviewer BjpG, the paper follows the state-of-the-art evaluation protocols for evasion attacks (e.g., adaptive attacks in the inductive setting). Several reviewers praised the comprehensiveness of the experimental evaluation. In the rebuttal, the authors provided additional experimental results which further strengthen the claim that the proposed approach improves robustness. However, I agree with Reviewer BjpG that the authors need to be more careful with claims regarding the accuracy vs. robustness trade-off, and carefully revise the paper to avoid over-claiming.

In my opinion, the issues raised by Reviewer ShLW who recommends a weak reject have been sufficiently addressed in the rebuttal, even though the reviewer did not update their score. Specifically, addressing baselines the authors explained again the validity-related limitations of adversarial training and why they only compare to GPRGNN. They also added a comparison to S2GC which underperforms in an inductive setup. The authors also added an additional experiment on the larger OGBN-Products dataset, although only using the DICE attack, claiming memory issues. I suggest that the authors include such larger scale experiments in the final version using scalable attacks followiing the setup in [1] which also report result on OGBN-Products using only 32 GB. Finally, the authors explained that the computational overhead is not significant.

The main remaining issue raised by Reviewer BFFi who also recommends rejection was novelty. However, after the rebuttal the reviewer holds a neutral position. In my opinion there is sufficient novelty. Still, the connections to test-time graph transformation raised by the reviewer are valid and I encourage the authors to include this discussion in the updated version of the paper.

Since Theorem 3.2 relies on having a Lipschitz constant L < 1, the authors should at least attempt to empirically verify whether this condition holds. Moreover, they should discuss what are the implications for their method (in theory and in practice) when this condition does not hold. While the loss indeed may encourages the condition, it is not clear whether it is actually enforced in practice.

Taking everything into consideration, I recommend the paper to be accepted.

References:
1. Geisler et al. "Robustness of graph neural networks at scale"